# Hyodeoxycholic acid ameliorates nonalcoholic fatty liver disease by inhibiting RAN-mediated PPARα nucleus-cytoplasm shuttling

Jing Zhong [1,2,9], Xiaofang He[1,9], Xinxin Gao[1], Qiaohong Liu[3], Yu Zhao [3], Ying Hong[1], Weize Zhu[1], Juan Yan[1], Yifan Li[1], Yan Li[1], Ningning Zheng[1], Yiyang Bao[1], Hao Wang[1], Junli Ma[1], Wenjin Huang[1], Zekun Liu[1], Yuanzhi Lyu [4], Xisong Ke[5], Wei Jia [6,7], Cen Xie [8]✉, Yiyang Hu [3]✉, Lili Sheng [1]✉ & Houkai Li [1]✉

Nonalcoholic fatty liver disease (NAFLD) is usually characterized with disrupted bile acid (BA) homeostasis. However, the exact role of certain BA in NAFLD is poorly understood. Here we show levels of serum hyodeoxycholic acid (HDCA) decrease in both NAFLD patients and mice, as well as in liver and intestinal contents of NAFLD mice compared to their healthy counterparts. Serum HDCA is also inversely correlated with NAFLD severity. Dietary HDCA supplementation ameliorates diet-induced NAFLD in male wild type mice by activating fatty acid oxidation in hepatic peroxisome proliferator-activated receptor α (PPARα)-dependent way because the anti-NAFLD effect of HDCA is abolished in hepatocyte-specific *Pparα* knockout mice. Mechanistically, HDCA facilitates nuclear localization of PPARα by directly interacting with RAN protein. This interaction disrupts the formation of RAN/CRM1/PPARα nucleus-cytoplasm shuttling heterotrimer. Our results demonstrate the therapeutic potential of HDCA for NAFLD and provide new insights of BAs on regulating fatty acid metabolism.

Nonalcoholic fatty liver disease (NAFLD) is the most common chronic liver disease in clinic with a spectrum of disorders ranging from simple fatty liver, nonalcoholic steatohepatitis, and fibrosis/cirrhosis. Currently, NAFLD affects nearly a quarter of the global population, and its prevalence is surging[1-3]. Emerging evidence has indicated the dynamic alterations in bile acid (BA) profiles throughout NAFLD progression[4,5]. Besides as digestive detergents, BAs are important signaling molecules to regulate lipid metabolism, glucose homeostasis, and immune response through acting on their receptors such as G protein-coupled bile acid receptor 1 (GBPAR1 or TGR5) and farnesoid X receptor

[1]School of Pharmacy, Shanghai University of Traditional Chinese Medicine, Shanghai 201203, China. [2]Huzhou Key Laboratory of Precision Medicine Research and Translation for Infectious Diseases, Affiliated Huzhou Hospital, Zhejiang University School of Medicine, Huzhou 313000, China. [3]Key Laboratory of Liver and Kidney Diseases (Ministry of Education), Institute of Liver Diseases, Shuguang Hospital Affiliated to Shanghai University of Traditional Chinese Medicine, Shanghai 201203, China. [4]Department of Dermatology, School of Medicine, University of California, Davis, Sacramento, CA, USA. [5]Institute of Interdisciplinary Integrative Medicine Research, Shanghai University of Traditional Chinese Medicine, Shanghai 201203, China. [6]Center for Translational Medicine, Shanghai Sixth People's Hospital Affiliated to Shanghai Jiao Tong University School of Medicine, Shanghai 200233, China. [7]School of Chinese Medicine, Hong Kong Baptist University, Kowloon Tong Hong Kong 999077, China. [8]State Key Laboratory of Drug Research, Shanghai Institute of Materia Medica, Chinese Academy of Sciences, Shanghai 201203, China. [9]These authors contributed equally: Jing Zhong, Xiaofang He. ✉e-mail: xiecen@simm.ac.cn; yyhuliver@163.com; llsheng@shutcm.edu.cn; hk_li@shutcm.edu.cn

(FXR)[6–8]. However, the role of specific BAs in NAFLD remains to be fully elucidated.

Hyocholic acid (HCA) species, including HCA, HDCA (hyodeoxycholic acid), and their glycine- and taurine-conjugated derivatives (GHCA, THCA, GHDCA, and THDCA), are the major BA species in pigs and are also present in humans and rodents[9]. Recent studies indicate HCA species, including HDCA, which are reduced in patients with obesity and diabetes, improve glucose homeostasis through a distinct TGR5 and FXR signaling mechanism[10–12]. In addition, HDCA is effective in attenuating gallstone formation, hyperlipidemia, and atherosclerosis[13–16]. However, the status and role of HDCA on NAFLD development is not clear.

The peroxisome proliferator-activated receptors (PPARs) are transcriptional regulators with crucial roles in metabolic disorders[17]. PPARα is predominantly expressed in hepatocytes in regulating fatty acid transport, fatty acid oxidation (FAO), and ketogenesis via activating the transcription of target genes in nucleus[18]. Hence, facilitating the nuclear localization of PPARα is a promising strategy against NAFLD. The intracellular content and function of PPARα rely on the transition balance between the cytoplasm and nucleus, which is orchestrated by the *ras* family GTPase RAN and the export receptor CRM1[19–22]. To date, there is little evidence of the interplay between RAN/CRM1/PPARα shuttling process and BAs functions on metabolic disorders.

In this study, we observed lower level of serum HDCA in NAFLD patients than healthy subjects, which inversely correlated with NAFLD severity, and was consistently reduced in liver and intestinal content of NAFLD mice. Supplementation of HDCA attenuated NAFLD in mice by facilitating the nuclear localization of PPARα, leading to enhanced hepatic fatty acid oxidation. Notably, the anti-NAFLD effect of HDCA was abolished in global and hepatocyte-specific *Pparα* knockout mice. HDCA hindered the formation of RAN/CRM1/PPARα export heterotrimer by direct binding with RAN protein, leading to the accumulation of nuclear PPARα. Our finding demonstrates that HDCA is a promising therapeutic agent for NAFLD and provides a therapeutic strategy for NAFLD by targeting the PPARα shuttling mechanism.

## Results

### Serum HDCA is reduced in patients with NAFLD

To investigate the serum BA composition and the changes of HCA species in patients with NAFLD, we recruited a cohort of 58 participants, which included 24 healthy and 34 NAFLD individuals (Fig. 1a). The clinical indices are shown in Supplementary Table 1. Compared with healthy controls, the NAFLD group had lower levels of HCA species, HDCA, GHCA, and THCA (Fig. 1b–h and S1). Spearman correlation analysis demonstrated the inverse correlations of HDCA and GHCA with clinical parameters of NAFLD (Fig. 1i). Further scatter plots showed that serum HDCA and GHCA levels, but not HCA level, were negatively associated with the controlled attenuation parameter (CAP) score, serum alanine transaminase (ALT) and aspartate transaminase (AST) levels (Fig. 1j and S1a). Additionally, the level of total HCA species was consistently reduced in the serum, liver, and different parts of intestinal content in NAFLD mice, which was induced by high fat and high sugar feeding (HFHS, high-fat diet with 30% sucrose in water) for 24 weeks (Fig. 1k, S2, S3). Among them, HDCA and its conjugated form THDCA were consistently decreased (Fig. 1l, m and S3), whereas HDCA was the only BA that consistently reduced in the serum of NAFLD patients and mice. Furthermore, HDCA was negatively correlated with liver weight, hepatic triglycerides (TG), serum ALT, and AST (Fig. 1n). These results demonstrated the inverse correlation between serum HDCA level and hepatic steatosis both in humans and mice, implying dietary supplementation of HDCA might have a protective effect against NAFLD.

### HDCA supplementation ameliorates NAFLD in mice

To test whether dietary supplementation of HDCA could attenuate NAFLD development, the anti-NAFLD effect of HDCA was examined in an HFHS-induced NAFLD mouse model. The results showed that HDCA reduced body weight and liver weight, especially attenuated hepatic steatosis and hepatic TG level in a dose-dependent manner (Fig. S4). Because the medium dose of HDCA (0.625% HDCA in diet) was effective in reducing lipid accumulation and inflammation in the liver, we adopted this dose in the following study and noticed accumulation of HDCA and its conjugated form THDCA, which were the highest elevated BAs, in mouse serum and liver after HDCA intervention (Fig. S5). HDCA supplementation did not affect total energy intake, but significantly reduced body weight, liver weight, and attenuated hepatic steatosis and inflammation scores, as well as NAFLD activity score (NAS) (Fig. 2a–e). However, typical ballooning hepatocytes were rarely found in this NAFLD model which might be due to the modeling method and duration as well as the genetic background of mice[23]. HDCA supplementation also reduced liver TG, serum ALT, and AST level, reversed HFHS-induced high expression of hepatic inflammation factors (Fig. 2f, g), and improved glucose homeostasis (Fig. 2h-i), implying the substantial benefits of HDCA on ameliorating metabolic disorders induced by HFHS diet. Meanwhile, the results showed that HDCA supplementation had minor impacts on the status of either apoptosis or fibrosis, which were also not obvious after 12-week HFHS feeding (Fig. S6a-c). Consistent with previous reports on decreased intestinal cholesterol absorption by HDCA treatment[13–15], our results revealed increased content of fecal cholesterol by HDCA intervention (Fig. S6d). However, the levels of fecal TG and non-esterified fatty acids (NEFA) were comparable between HFHS and HDCA groups (Fig. S6d). Consistently, both BODIPY and Oil Red O staining of the jejunum indicated that HDCA did not affect lipid absorption (Fig. S6e), suggesting that liver TG reduction in HDCA-treated mice is not due to suppression on intestinal lipid absorption. In addition, similar effects of HDCA were also observed in *ob/ob* mouse model, including reduced liver weight, liver TG, hepatic steatosis, and serum ALT and AST levels (Fig. S7). Interestingly, HDCA had a minor impact on chow diet-fed C57BL/6 J mice (Fig. S8). Collectively, these data suggested that HDCA was effective in ameliorating hepatic steatosis and/or inflammation in diet- and genetic-induced NAFLD mouse models.

### HDCA activates fatty acid oxidation pathway in hepatocytes

In an attempt to understand the molecular mechanism underlying the protective effect of HDCA, proteomic profiles of mouse liver tissue were analyzed utilizing Tandem Mass Tag-based quantitative proteomic approach. A total of 1190 and 623 proteins were significantly changed in control *vs.* HFHS group and in HFHS *vs.* HDCA group (*p* value < 0.05 and fold change >1.2), respectively (Fig. 3a and S9). Of the 430 proteins that overlapped, 420 proteins were reversed by HDCA (Fig. 3a). KEGG pathway analysis of the 420 differential proteins demonstrated that among the highly enriched pathways, metabolic pathways, oxidative phosphorylation, NAFLD, and PPAR signaling pathway were closely associated with hepatic fatty acid metabolism (Fig. 3b). Heatmap of fatty acid metabolism-related proteins showed that most FAO associated proteins were downregulated by HFHS and upregulated by HDCA, implying the primary impact of HDCA on FAO process (Fig. 3c).

In addition, hepatic transcriptomics was also adopted by using RNA-seq analysis. A total of 2039 and 1121 genes differed significantly in control *vs.* HFHS group and in HFHS *vs.* HDCA group (*p* value < 0.05 and fold change >2), respectively (Fig. 3d). Among the 769 overlapped genes, 762 genes were reversed by HDCA (Fig. 3d). Based on the genes that were upregulated by HFHS and downregulated by HDCA, the significantly enriched KEGG pathways included many inflammation and infection-related pathways such as cytokine-cytokine receptor

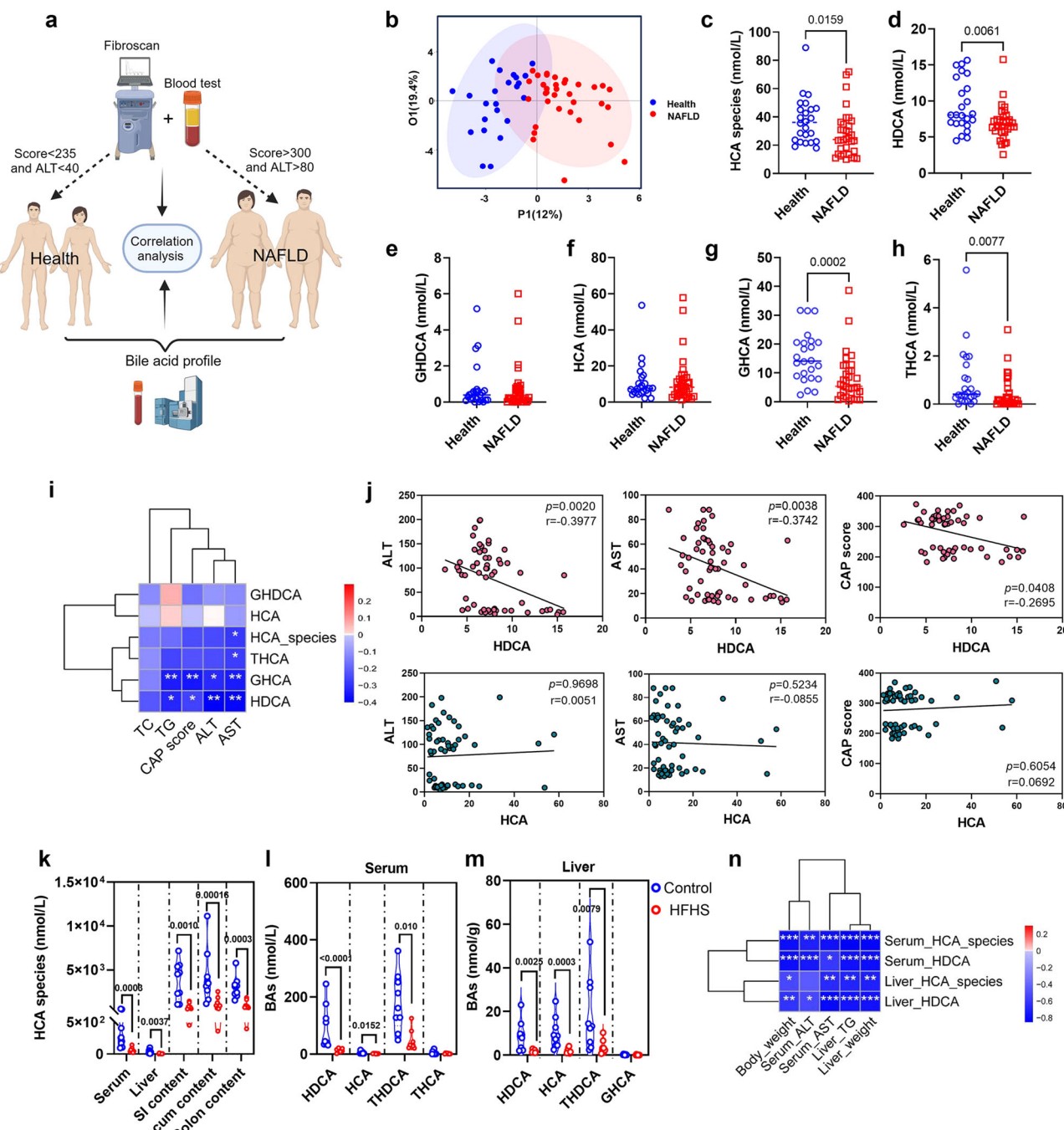

**Fig. 1 | The level of HDCA is reduced in patients with NAFLD and diet-induced NAFLD mice. a** Schematic representation of the clinical sample analysis. Created with BioRender.com. **b** OPLSDA plot of serum BA profile in health ($n = 24$) and NAFLD ($n = 34$) individuals. **c–h** Serum level of HCA species, HDCA, GHDCA, HCA, GHCA, and THCA ($n = 24$ for Health, $n = 34$ for NAFLD). **i** Spearman correlation analysis between HCA species BAs and clinical characteristics (*$p < 0.05$, **$p < 0.01$). **j** Scatter plots of serum HDCA and HCA with serum ALT, AST, and CAP score. **k–n** The C57BL/6 mice were fed an HFHS diet for 24 weeks. **k** Levels of HCA species in the serum, liver, small intestinal (SI) content, cecum content, and colon content

of diet-induced mice with NAFLD ($n = 9$ for Control, $n = 8$ for HFHS). **l** Level of individual HCA species in the serum of NAFLD mice ($n = 9$ for Control, $n = 8$ for HFHS). **m** Level of individual HCA species in the liver of NAFLD mice ($n = 9$ for Control, $n = 8$ for HFHS). **n** Spearman correlation analysis between HCA species, HDCA, and mouse phenotypes (*$p < 0.05$, ** $p < 0.01$, ***$p < 0.001$). Data are presented as mean values ± SEM. Differences between groups were determined by unpaired two-tailed Mann–Whitney $U$ test. Source data are provided as a Source Data file.

interaction and chemokine signaling pathway, which were further confirmed by GSEA results (Fig. 3e, f and S10). Based on the genes that were downregulated by HFHS but upregulated by HDCA, the enriched KEGG pathways included retinol metabolism, steroid hormone biosynthesis, fatty acid degradation, and PPAR signaling pathway (Fig. 3g and S10). GSEA results also revealed the enrichment of fatty acid

oxidation (11 core enriched genes out of 23 genes) and PPARα targets (11 core enriched genes out of 15 genes) in HDCA group (Fig. 3h).

Mitochondrial fatty acid β-oxidation is the critical step of fatty acid metabolism. Our results showed HDCA significantly inhibited the expression of fatty acid uptake protein CD36, but activated the expression of intracellular fatty acid binding protein 1 (FABP1) and

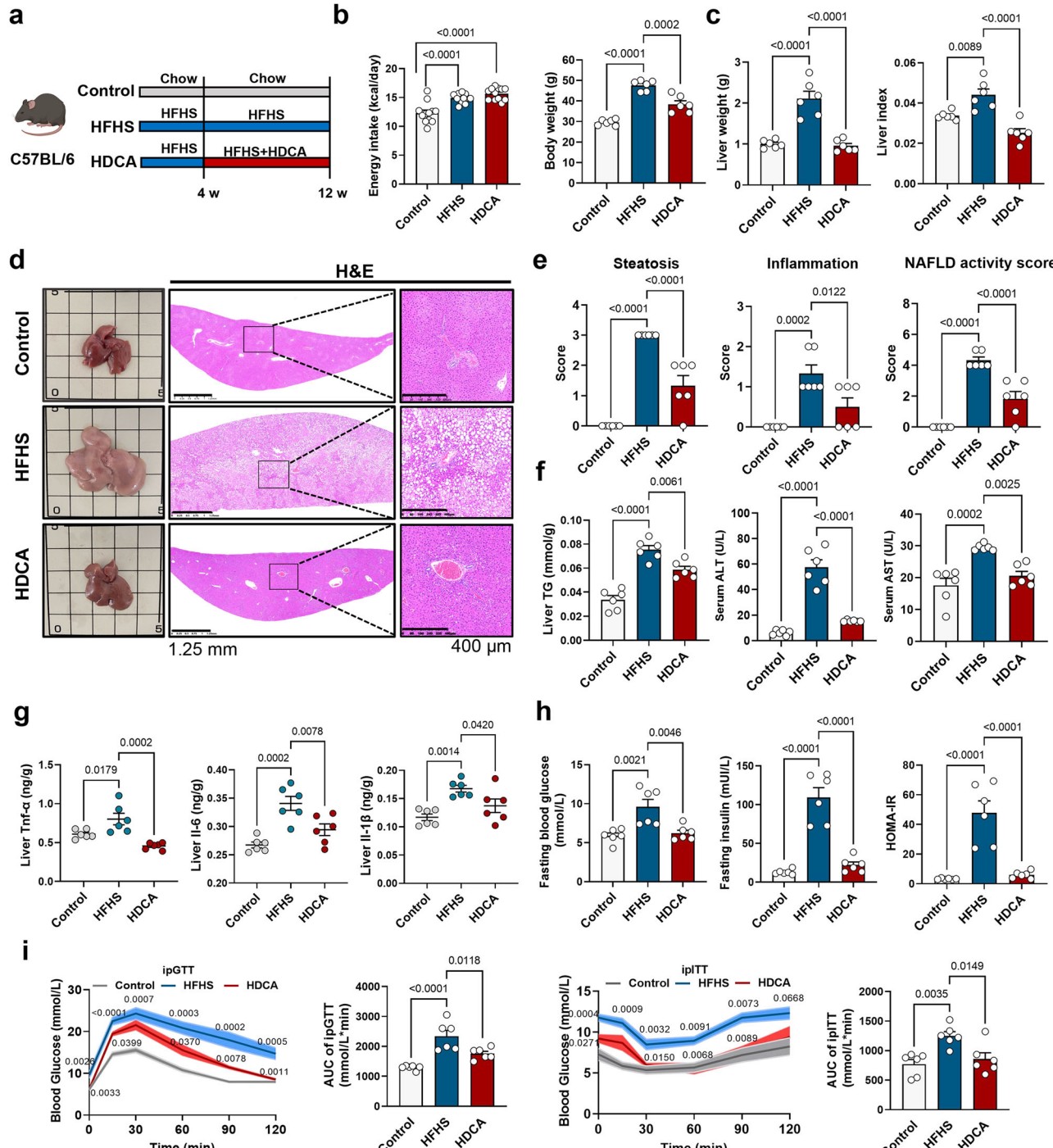

**Fig. 2 | HDCA supplementation ameliorates NAFLD in mice.** The C57BL/6 mice in control group and HFHS group were fed with chow or HFHS diet for 12 weeks. HDCA group were fed with HFHS for 4 weeks and then supplemented with 0.625% HDCA in the HFHS diet for another 8 weeks. **a** Schematic representation of HDCA intervention in HFHS-fed C57BL/6 mice (*n* = 6 per group). Mouse element created with BioRender.com. **b** Energy intake (energy intake was recorded every week for 12 weeks, *n* = 12 per group) and body weight (*n* = 6 per group). **c** Liver weight and liver index. **d** Representative images of liver general appearance, H&E staining (Scale bar, 1.25 mm and 400 µm). **e** Steatosis score, inflammation score, and NAFLD activity score. **f** Liver TG, serum ALT and AST level. **g** Levels of TNF-α, IL-6, and IL-1β in the liver. **h** Fasting blood glucose, fasting insulin, and HOMA-IR. **i** Intraperitoneal glucose tolerance test (ipGTT) and intraperitoneal insulin tolerance test (ipITT) results with area under the curve (AUC) calculation. **c**–**i** *n* = 6 per group. Data are presented as mean values ± SEM. Difference between groups were determined by one-way ANOVA test followed by Tukey's multiple comparison. **i** *p* values around the blue line represent comparison with control group, the *p* values around the red line represent comparison with HFHS group. Source data are provided as a Source Data file.

carnitine palmitoyltransferase 2 (CPT2) at both protein and mRNA levels, leading to increased transportation of fatty acid to the mitochondria (Fig. 3i). Of the 7 key β oxidation enzymes, the expression of 5 enzymes including ACADL, ACADVL, HADHA, HADHB, and ACAA2

were significantly increased by HDCA at both protein and mRNA levels, suggesting the elevation of fatty acid β oxidation. In addition, increased ketogenesis is a manifestation of increased FAO. We found the expression of HMGCS2, the rate-limiting enzyme for ketogenesis,

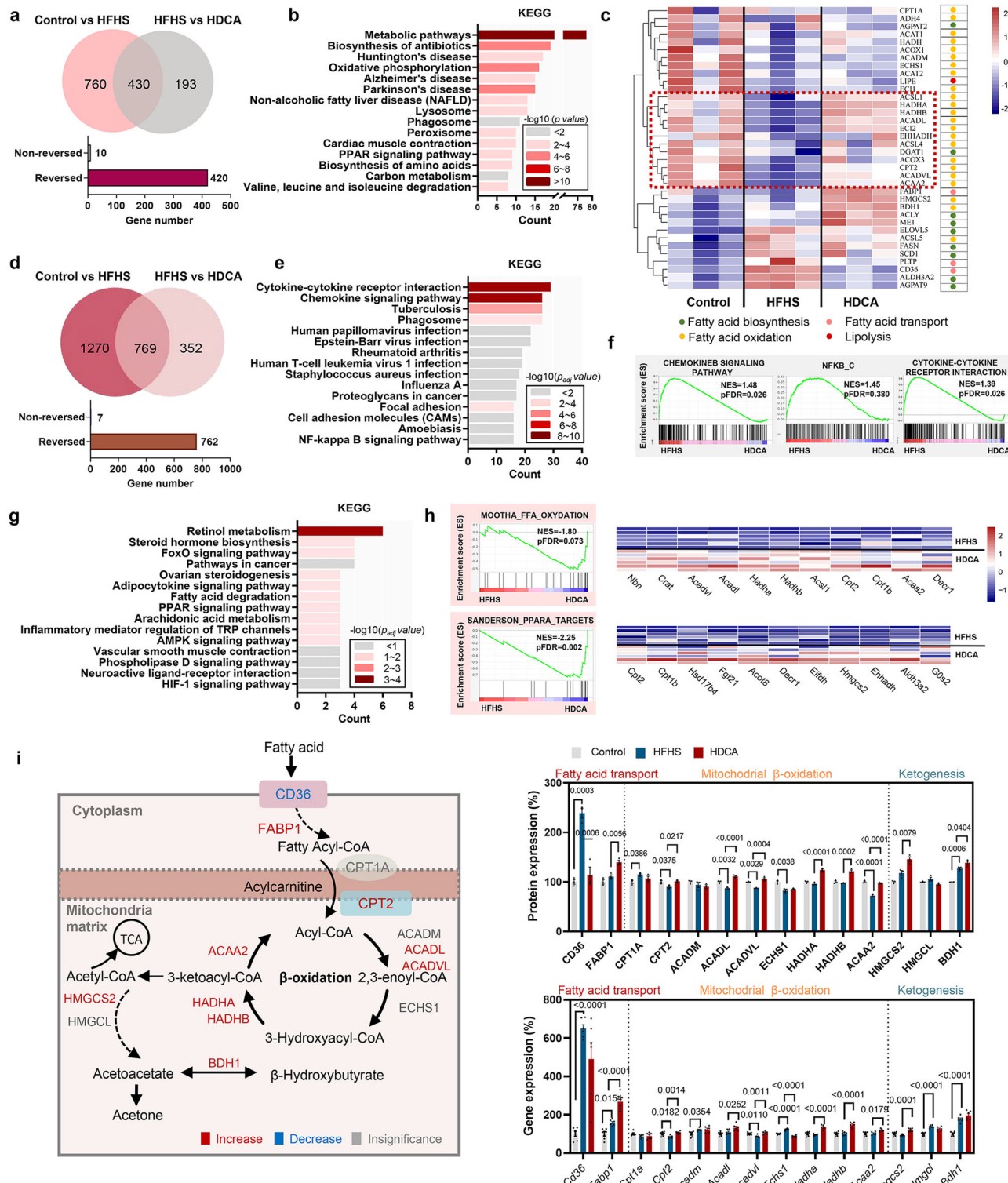

**Fig. 3 | HDCA activates fatty acid oxidation in hepatocytes. a** Venn diagram of significantly regulated proteins by HFHS or by HDCA as shown by proteomics. **b** Top 15 pathways enriched with KEGG enrichment analysis based on the 420 reserved proteins (*n* = 3 per group). **c** Heatmap of proteins related to fatty acid metabolism. **d** Venn diagram of significantly regulated genes by HFHS or by HDCA as shown by transcriptomics (*n* = 6 per group). **e** Top 15 pathways with KEGG enrichment analysis based on the genes that are upregulated by HFHS and down-regulated by HDCA and (**f**) GSEA analysis of chemokine signaling pathway, NFκB, and cytokine-cytokine receptor interaction. **g** Top 15 pathways with KEGG enrichment analysis based on the genes that are downregulated by HFHS and upregulated

by HDCA and **h** GSEA analysis of fatty acid oxidation and PPARα targets. **i** Schematic diagram of fatty acid transport, β-oxidation, and ketogenesis in the mitochondria, and the expression changes of related proteins and genes in this process (Red and blue represent upregulation and downregulation of proteins and genes in HDCA group compared with HFHS group, respectively. Gray represents no significate difference between two groups) (*n* = 3 per group). Data are presented as mean values ± SEM. Difference between groups were determined by one-way ANOVA test followed by Tukey's multiple comparison. The average of gene and protein expression in control group is normalized as 100%. Source data are provided as a Source Data file.

was upregulated by HDCA intervention. Moreover, the level of β-hydroxybutyrate, which derives from FAO, was significantly increased in serum of mice in HDCA group than HFHS group (Fig. S11). These data suggested that the anti-NAFLD effect of HDCA was mainly associated with the activation of FAO in hepatocytes.

## The anti-NAFLD effect of HDCA is hepatic PPARα-dependent

PPARα is the key regulator for FAO by triggering the transcription of genes involved in FAO within the nucleus. Interestingly, HDCA did not alter the total content of PPARα protein, but increased its nuclear to cytoplasmic ratio that was reduced by HFHS feeding (Fig. 4a, b). Consistent with the proteomics data, the protein levels of PPARα targets, such as FABP1, CPT2, and HMGCS2, were significantly increased, while the expression of CD36 was reduced in HDCA group (Fig. 4c). Additionally, the expression of many other PPARα targets were increased in HDCA group, as revealed by proteomics and transcriptomics data (Fig. S12). Besides its role in FAO, HDCA also attenuated inflammation (Fig. 4d, e). To test whether HDCA can directly regulate PPARα nuclear localization, AML12 cells were treated with HDCA at two dosages. The results showed HDCA dose-dependently increased the nuclear content of PPARα protein without affecting its total expression, as well as increased expression of CPT1, CPT2, and FABP1 proteins (Fig. 4f). Nevertheless, THDCA, which is also elevated after HDCA intervention, had no effect on regulating PPARα subcellular localization (Fig. S13). Moreover, HDCA reduced the expression of chemokines and cytokines in AML12 cells and RAW264.7 macrophages, suggesting the anti-inflammation effect of HDCA (Fig. 4g, h). Therefore, both in vivo and in vitro evidence demonstrated that HDCA promoted PPARα nuclear localization, and thus activated PPARα-mediated FAO and inhibited inflammation.

To further explore whether the anti-NAFLD effect of HDCA was PPARα-dependent, the whole body *Pparα* knockout (*Pparα*[−/−]) mice were constructed (Fig. S14a–c). In comparison with the findings on wild-type mice (Fig. 2), HDCA intervention failed to improve NAFLD-related parameters in *Pparα*[−/−] mice such as body weight, liver TG, hepatic steatosis score, and NAS (Fig. 5a–f). Notably, HDCA had no impact on the expression of FABP1, CPT2, and HMGCS2 in *Pparα*[−/−] mice (Fig. 5g). In addition, the contents of hepatic TNF-α, IL-6, and IL-1β were not affected by HDCA supplementation in *Pparα*[−/−] mice (Fig. 5h). Collectively, these results demonstrated that whole body PPARα deficiency abolished the anti-NAFLD effect of HDCA.

To further determine the specific role of hepatic PPARα in response to HDCA, *Pparα*[flox/flox] mice were administered AAV2/8-TBG-Cre or control virus AAV2/8-TBG-ZsGreen to obtain hepatocyte-specific PPARα deficiency mice (*Pparα*[hep−/−]) or control mice (Fig. S14d-g). Similar to the response of *Pparα*[−/−] mice, the liver TG, hepatic steatosis score, inflammation score, and NAS of *Pparα*[hep−/−] mice did not show any changes after HDCA supplementation (Fig. 5i-n). Consistently, the enhanced expression of FABP1, CPT2, HMGCS2 proteins, and reduced levels of inflammation cytokines induced by HDCA in the liver of *Pparα*[flox/flox] mice were eliminated in hepatocyte-specific PPARα deficient mice (Fig. 5o-p). Therefore, these results suggested that the anti-NAFLD effect of HDCA was dependent on modulation of liver PPARα.

## HDCA inhibits nuclear export of PPARα in an FXR-independent manner

Hepatic FXR is essentially involved in BAs-mediated lipid metabolism[6–8]. Previous study indicated HDCA inhibited FXR activity in the presence of FXR agonist (chenodeoxycholic acid)[11]. Our results revealed that HDCA intervention mildly activated FXR transcriptional activity in HEK293T cells by luciferase reporter gene assays and induced the mRNA expression of *Shp* gene in AML12 cells, respectively (Fig. 6a-b), implying HDCA alone was a weak agonist of FXR in HEK293T cells. This discrepancy might be due to direct activation of

FXR or other unidentified factors modulated by HDCA in AML12 cells. We further observed that HDCA consistently facilitated PPARα nuclear localization and upregulated the expression of PPARα targeted genes in the presence or absence of FXR antagonist, Guggulsterone, in AML12 cells (Fig. 6b-c), suggesting the effect of HDCA on facilitating PPARα localization was FXR independent.

The nucleus-cytoplasm shuttling of PPARα is tightly controlled by importins and exportins[19,22]. Inhibition of nuclear import receptor by importazole (IPZ) caused cytoplasmic PPARα accumulation, whereas HDCA still facilitated nuclear PPARα accumulation in the presence of IPZ. Meanwhile, inhibition of nuclear export receptor with leptomycin B (LMB) induced nuclear accumulation, while combined treatment with LMB and HDCA did not further increase the amount of PPARα in nucleus (Fig. 6d). These findings suggested that HDCA might regulate the nuclear export process of PPARα, rather than nuclear import.

Two proteins, export receptor CRM1 and the small *ras* family GTPase RAN, play prominent roles in nuclear protein export by binding export cargo protein to form an export heterotrimer[19–22]. In AML12 cells, HDCA significantly increased nuclear CRM1 level but decreased cytoplasmic CRM1 level without affecting its total protein expression (Fig. 6e). Consistent with the in vitro result, the protein levels of nuclear and cytoplasmic CRM1 were increased and decreased, respectively in the liver of HDCA treated mice comparing to HFHS group (Fig. 6f). However, the expression or localization of RAN was not affected by HDCA (Fig. 6e, f). Together, these findings suggested HDCA promoted nuclear localization of PPARα by suppressing CRM1-mediated nuclear export process.

## HDCA directly binds with RAN protein to inhibit the formation of nucleus-cytoplasm shuttling heterotrimer, RAN/CRM1/PPARα

To elucidate the molecular mechanism behind the HDCA-mediated nuclear accumulation of PPARα and CRM1, we screened for potential HDCA-binding proteins that involved in nuclear transport by using biotin-labeled HDCA and HuProt human protein microarray (Fig. 7a). With the criteria of Z score >2.8 and I Mean ratio >1.4, we found HDCA preferably bound to RAN, but not CRM1 or PPARα (Fig. 7b and S15). Cellular thermal shift assay (CETSA) in AML12 cells further confirmed the increased thermal stability of RAN after HDCA treatment, suggesting the direct interaction of HDCA with RAN (Fig. 7c). Based on this finding, we hypothesized that HDCA reduced the formation of RAN/CRM1/PPARα heterotrimer by directly binding with RAN protein, leading to the blockage of PPARα nucleus-cytoplasm shuttling (Fig. 7d). We then performed Co-immunoprecipitation (Co-IP) assay to study their interactions in AML12 cells. The Co-IP results showed that RAN was readily detected in CRM1 immunoprecipitants, while the interaction between RAN and CRM1 was reduced by HDCA. Similarly, clear interaction between CRM1 and PPARα was observed, and HDCA reduced the binding of CRM1 to PPARα (Fig. 7e). We further verified these bindings in HEK293T cells by overexpressing these three proteins. Consistent with the observation in AML12 cells, the interactions of RAN/PPARα and RAN/CRM1 were disrupted by HDCA (Fig. 7f). Nevertheless, since there is endogenous expression of RAN in HEK293T cells, we also observed reduced CRM1/PPARα binding after HDCA treatment (Fig. 7f).

Proximity ligation assay (PLA) was further performed to validate the interplay of RAN/CRM1/PPARα in AML12 cells (Fig. 7g). The RAN/PPARα interaction was significantly disrupted in the presence of HDCA, though the signal was relatively weak because of the relatively far distance between RAN and PPARα. Consistent with Co-IP results, the interactions of RAN/CRM1 and CRM1/PPARα in the nucleus were disrupted upon HDCA treatment. Further molecular docking study between RAN and HDCA showed a binding free energy of −7.87 kcal/mol, anchored predominantly at Gly-5, Pro-49, and Val-51 (Fig. 7h). Mutation of these three amino acids almost abolished the effect of

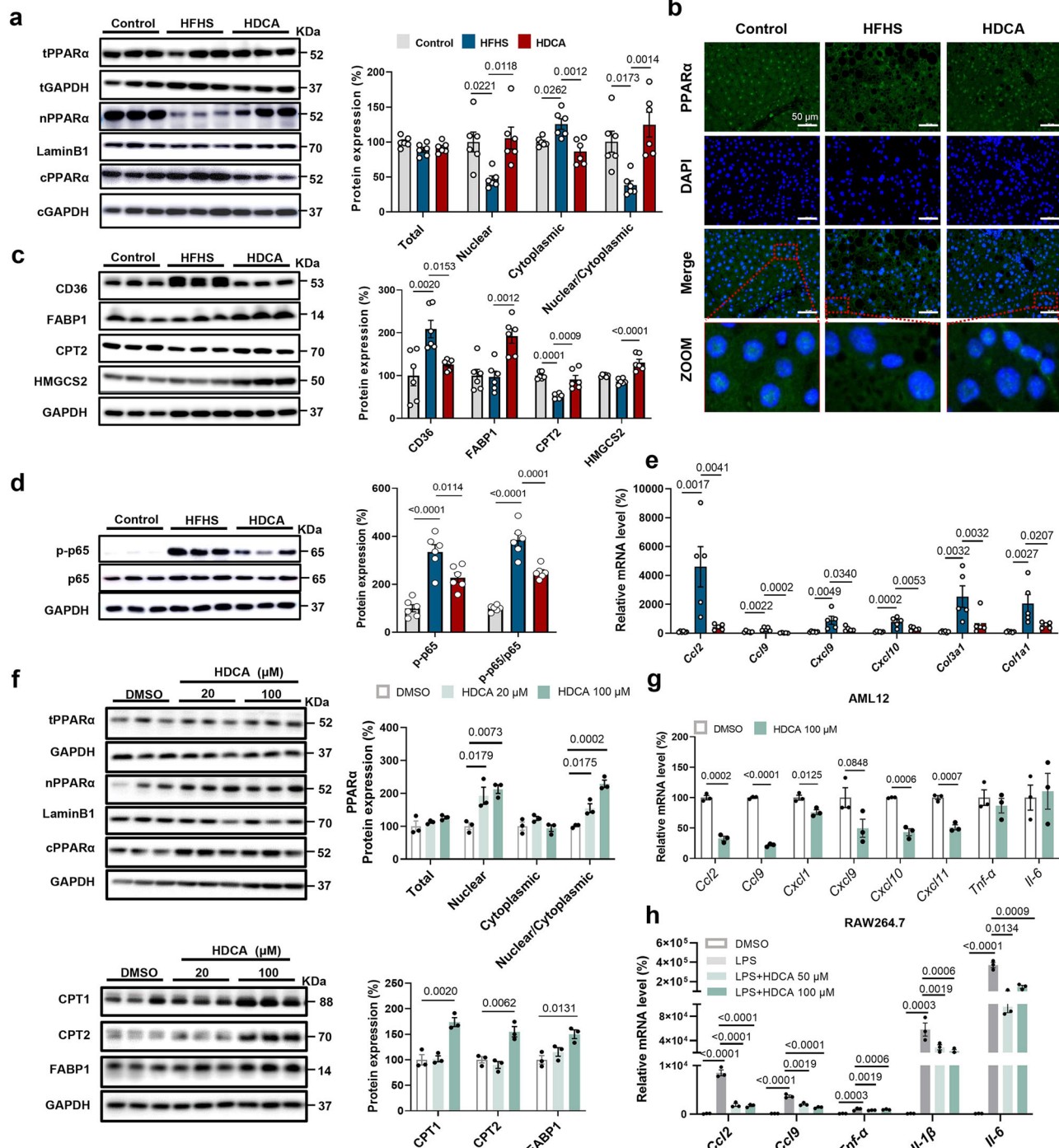

**Fig. 4 | HDCA stimulates nuclear accumulation of PPARα and promotes the expression of PPARα target gene. a–e** Liver samples are from the HFHS-fed C57BL/6 mice with 0.625% HDCA intervention as in Fig. 2a. **a** HDCA intervention promoted the nuclear accumulation of PPARα (*n* = 6 per group). **b** Immunofluorescence staining of liver PPARα, Scale bar, 50 μm. The immunofluorescence analyses were representative of data from three mice. **c, d** Protein expressions of CD36, FABP1, CPT2, HMGCS2, p-p65, p65, IKBα in the liver (*n* = 6 per group). **e** mRNA expressions of liver *Ccl2, Ccl9, Cxcl9, Cxcl10, Col3a1, Col1a1* (*n* = 6 for Control, *n* = 5 for HFHS and HDCA). **f** Protein expressions of PPARα at 24 h, and protein expressions of CPT1, CPT2, FABP1 at 48 h in AML12 cells treated with or without 20 μM and 100 μM HDCA (*n* = 3 per group). **g** mRNA expressions of

inflammation-related genes in AML12 cells after 100 μM HDCA treatment for 24 h (*n* = 3 per group). Difference between groups were determined by two-tailed Student's *t* test. **h** mRNA expressions of *Ccl2, Ccl9, Tnf-α, Il-1β, Il-6* in RAW264.7 macrophages (*n* = 3 per group). The RAW264.7 cells were treated with HDCA (50, 100 μM) for 24 h, in the last 6 hours of this experiment, LPS (20 ng/ml) was added into the medium. Except (**g**), difference between groups determined by one-way ANOVA test followed by Tukey's multiple comparison. The findings in **f–h** were confirmed in three independent experiments. Data are presented as mean values ± SEM. The average of gene and protein expression in control group is normalized as 100%. Source data are provided as a Source Data file.

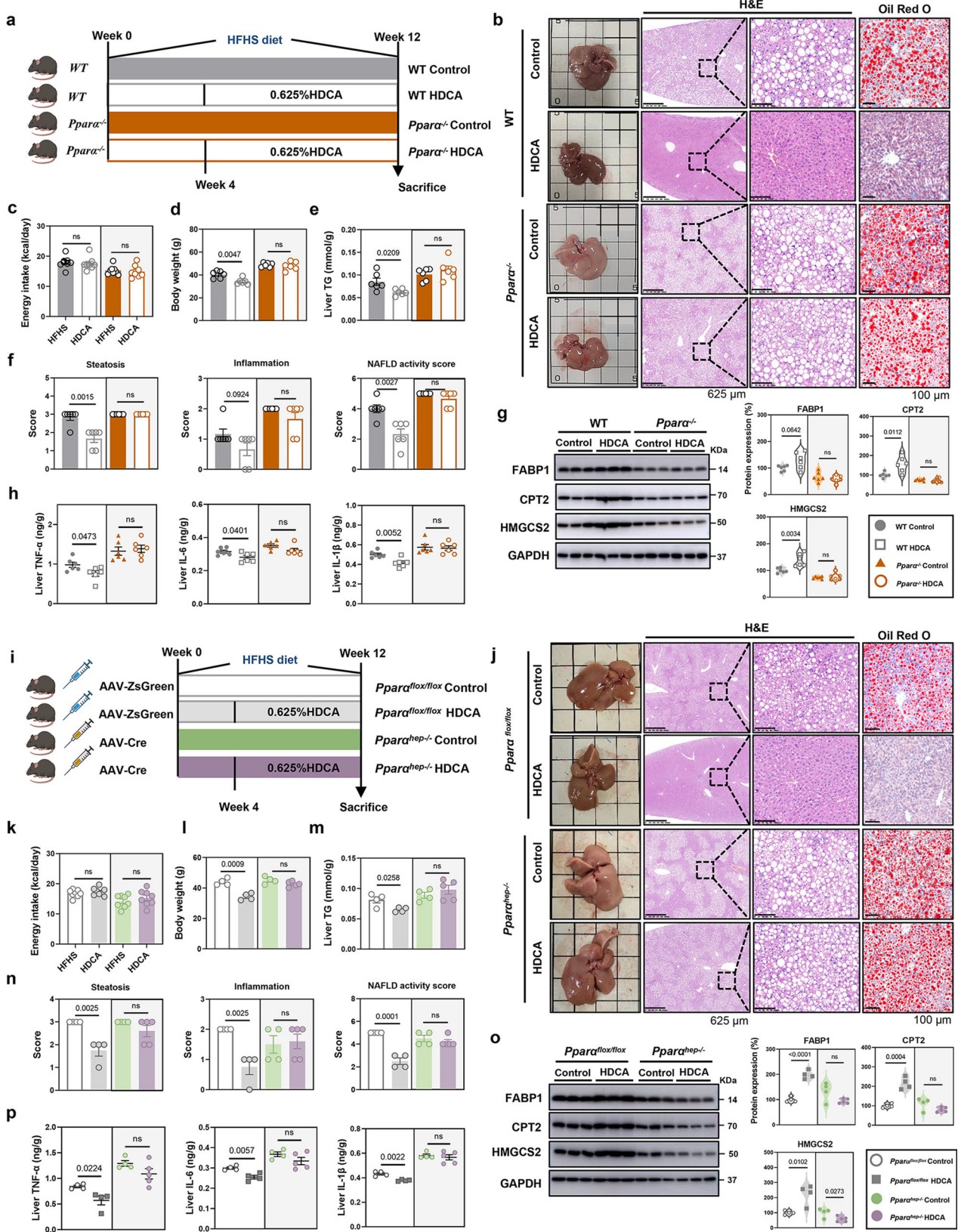

HDCA on reducing RAN/CRM1 interaction (density values were 1 *vs.* 0.67 and 1 *vs.* 0.58 in wild type RAN as shown in Fig. 7e, f, and 1 *vs.* 0.93 after RAN mutation as shown in Fig. 7i based on Western blot), suggesting Gly-5, Pro-49, and Val-51 might play essential role in the binding of HDCA with RAN. Altogether, these data suggested that HDCA could bind to RAN to reduce the formation of nucleus-cytoplasm shuttling heterotrimer, RAN/CRM1/PPARα, leading to increased nuclear accumulation of PPARα.

## Discussion
BAs play critical roles in regulating lipid and glucose homeostasis, as well as inflammation[6–8]. Mounting evidence revealed the close

**Fig. 5 | PPARα deficiency abolishes HDCA-mediated anti-NAFLD effect.**
**a–h** Control groups of C57BL/6 wild-type mice and *Pparα*−/−/−  mice were fed with HFHS for 12 weeks. HDCA groups of C57BL/6 wild-type mice and *Pparα*−/− mice were fed with HFHS for 4 weeks and then supplemented with 0.625% HDCA in the diet for another 8 weeks. **a** Schematic of HDCA intervention in HFHS-fed C57BL/6 wild type mice and *Pparα*−/− mice (*n* = 6 per group). Mouse element created with BioRender.com. **b** Representative images of liver general appearance, H&E staining (Scale bar, 625 μm and 100 μm), and Oil Red O staining (Scale bar, 100 μm). **c–e** Energy intake, body weight, and liver TG. **f** Steatosis score, inflammation score, and NAFLD activity score. **g** Protein expressions of liver FABP1, CPT2, and HMGCS2. **h** TNF-α, IL-6, IL-1β levels in the liver. **i–p** Control groups of *Pparα*flox/flox and *Pparα*hep−/− mice were fed with HFHS for 12 weeks. HDCA groups of *Pparα*flox/flox and *Pparα*hep−/− mice were fed with HFHS for 4 weeks and then supplemented with 0.625% HDCA in the diet for another 8 weeks. **i** Schematic of HDCA intervention in HFHS-fed *Pparα*flox/flox and *Pparα*hep−/− mice (*n* = 4 for *Pparα*flox/flox Control /HDCA and *Pparα*hep−/−Control, *n* = 5 for *Pparα*hep−/−HDCA). Mouse and syringe elements created with BioRender.com. **j** Representative images of liver general appearance, H&E staining (Scale bar, 100 μm and 625 μm) and Oil Red O staining (×400). **k–m** Energy intake, body weight, and liver TG. **n** Steatosis score, inflammation score, and NAFLD activity score. **o** Protein expressions of liver FABP1, CPT2, and HMGCS2. **p** TNF-α, IL-6, IL-1β levels in the liver. Data are presented as mean values ± SEM. Difference between groups were determined by two-tailed Student's *t* test. The average of protein expression in control group is normalized as 100%. Source data are provided as a Source Data file.

relationship between the changes of BAs and NAFLD development[4]. HCA species, including HDCA, are decreased in diabetic subjects[12] and HDCA administration can improve glucose homeostasis by regulating BA receptor TGR5 and/or FXR in a GLP-1 dependent manner[10,11]. However, the correlation of HCA species with NAFLD is not clear. In our current study, we found HCA species, especially HDCA, were consistently reduced in patients and mice with NAFLD and negatively associated with fatty liver characteristics. Similar change was observed in HFD-fed rats with hepatic steatosis[24]. Interestingly, the reduction fold of HCA species was less in humans than in mice, and GHCA, which was significantly reduced in serum of NAFLD patients, was not detected in mouse serum or different in the liver of NAFLD mice. These discrepancies of BA changes between species might be attributed to the differences in the synthesis (including conversion of unconjugated to glycine- or taurine-conjugated BAs), transportation, and re-absorption of BAs, as well as gut microbiota-mediated secondary BAs production[25,26].

HDCA is well-known for its hypolipidemic effect and inhibition of atherosclerotic lesion formation by decreasing intestinal cholesterol absorption and increasing cholesterol efflux[13-15]. We found dietary supplementation of HDCA efficiently ameliorated HFHS diet-induced NAFLD and insulin resistance in mice. In addition, the ALT and AST levels were also dramatically reduced after HDCA treatment, suggesting the protective effect of HDCA on liver injury. Moreover, because of the comparable levels of fecal TG and NEFA as well as BODIPY and Oil Red O staining results between HFHS and HDCA groups, the reduction of liver TG accumulation in HDCA-treated mice was not due to less lipid absorption. To reveal the potential mechanism of the anti-NAFLD effect, proteomic and transcriptomic analysis of liver tissue were performed. The results consistently showed HDCA reversed HFHS-induced alteration of PPARα signaling pathway, suggesting PPARα, the master nuclear transcriptional factor that regulates FAO in liver, might play an essential role in the anti-NAFLD effect of HDCA. β-hydroxybutyrate is the main ketone body produced by the hepatocytes through FAO[27]. Consistent with previous study[28,29], HFHS diet increased serum β-hydroxybutyrate level, and HDCA further upregulated it. The divergent changes of nuclear PPARα, HMGCS2, and β-hydroxybutyrate level might due to other transcription factors and posttranslational modification of HMGCS2, and might be a compensatory response of the body. Furthermore, HDCA and THDCA are the highest increased BAs in the liver and serum of mice with HDCA intervention. Our results indicated that HDCA induced nuclear localization of PPARα both in vivo and in vitro, while THDCA has no such effect, suggesting HDCA, but not its conjugated form, facilitated the accumulation of nuclear PPARα.

PPARα is a key regulator in lipid metabolism and also plays a critical role in the inflammatory response[18,30]. Numerous data showed the expression of PPARα was altered during the development of NAFLD and NASH both in animal models and in patients[31-35]. The liver PPARα expression was negatively associated with severity of steatosis, NASH, and fibrosis[31]. Whole-body

knockout of *Pparα* or hepatocyte-specific deletion of *Pparα* promoted the development of NAFLD, suggesting *Pparα* is a potential drug target of NAFLD/NASH[36-38]. Indeed, great efforts have been invested in the discovery of PPARα agonists and several drugs targeting PPARα are under clinical study[39]. Our results also showed that the anti-NAFLD effect of HDCA was abolished in global *Pparα* knockout mice, implying that *Pparα* is crucial for the effect of HDCA. Notably, the function of PPARα is tissue-specific. Global *Pparα* knockout mice are susceptible to NAFLD development, but resistant to insulin resistance[34,40], while hepatocyte-specific *Pparα* knockout mice only partially phenocopy the phenotypes of global *Pparα* deletion[36-38,41]. In contrast, intestine-specific *Pparα* knockout mice are protected against obesity and NAFLD[42]. Thus, we constructed hepatocyte-specific *Pparα* knockout mice to investigate the role of hepatic PPARα in HDCA intervention. The results showed that the anti-NAFLD effect of HDCA was abolished by liver PPARα deletion. Therefore, our findings demonstrated the importance of liver PPARα in mediating the anti-NAFLD effect of HDCA in mice.

The expression of PPARα gene is tightly regulated by various physiological conditions including fasting, aging, and hormones[18]. Many drugs, natural compounds, and endogenous functional proteins have been reported to participate in hepatic fatty acid metabolism by regulating the expression of PPARα[43-45]. In addition to expression regulation, the activity of PPARα is also controlled by natural and synthetic ligands including fatty acids and their derivatives[18], post-translational modification including phosphorylation and ubiquitination[46-48], and cofactor recruitment[49,50]. As a nuclear transcription factor, nuclear localization is essential for PPARα function, which is determined by the dynamic shuttling balance between cytoplasm and nucleus. Previous studies have shown that the nucleus-cytoplasm shuttling of PPARα is ligand-dependent[51]. In addition, the ubiquitin ligase MuRF1 regulates cardiac PPARα by enhancing nuclear export via monoubiquitination, and MEK1 inhibits cardiac PPARα activity by direct interaction to stimulate PPARα export from the nucleus[52,53]. However, the precise regulation of PPARα subcellular localization in hepatocyte was not fully understood, especially during the HDCA intervention. In our study, we found that HDCA reduced PPARα nuclear export but not its import. After inhibition of FXR, HDCA still induced the expression of FAO-related genes and the effect of HDCA on PPARα subcellular localization was FXR-independent. Specifically, HDCA could directly bind with RAN at Gly-5, Pro-49, and Val-51, and consequently inhibited the binding of RAN with CRM1, resulting in less formation of RAN/CRM1/PPARα export heterotrimer. In this way, HDCA stimulated nuclear localization of PPARα and enhanced PPARα activity to regulate fatty acid oxidation (Fig. 8).

In conclusion, we uncover a mechanism underlying the anti-NAFLD effect of HDCA by facilitating PPARα nuclear localization via binding with RAN protein. Our study provides a therapeutic strategy for NAFLD by targeting PPARα nucleus-cytoplasm shuttling heterotrimer.

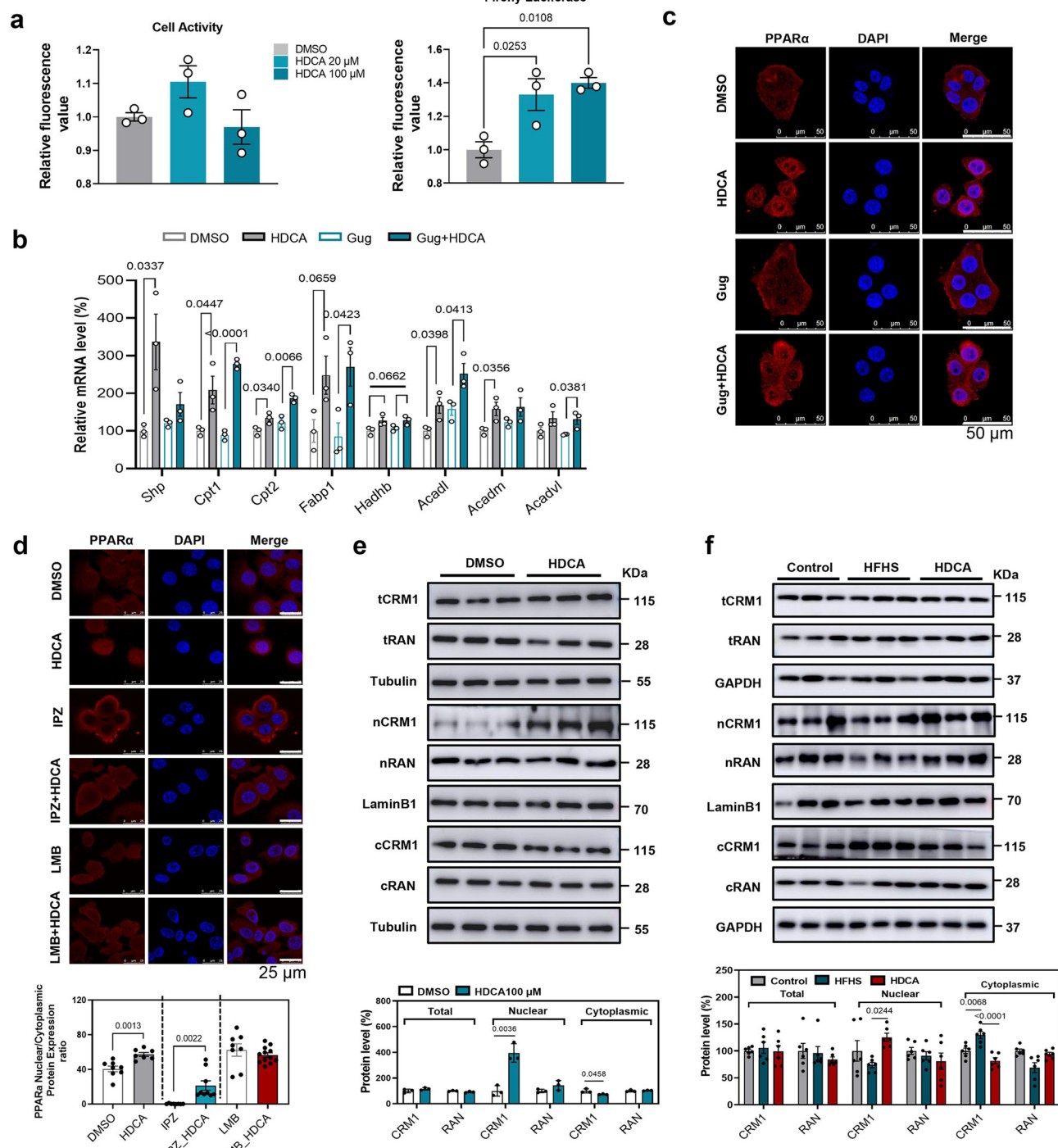

**Fig. 6 | HDCA inhibits nuclear export of PPARα in an FXR-independent manner.**
**a** Luciferase assay of HDCA on FXR activity in HEK293T cells (n = 3). After 24 h co-transfection with pCMV-Script-hFXR and PGL4-Shp-TK firefly luciferase plasmids, followed by continuous 6 h starving in 1% FBS DMEM medium, the HEK293T cells were exposed to DMSO or HDCA (20 μM and 100 μM) for 24 h to test FXR activity. The left panel was cell viability test, and the right panel was Luciferase assay on FXR activity. Difference between groups were determined by one-way ANOVA test followed by Tukey's multiple comparison. **b** mRNA expressions of FXR downstream and fatty acid oxidation related genes treated by FXR antagonist, Guggulsterone (20 μM) with or without HDCA (100 μM) for 24 h in AML12 cells (n = 3). Difference between groups were determined by unpaired two-tailed Student's t test. **c** Immunofluorescence of PPARα with Guggulsterone (20 μM) and/or HDCA (100 μM) treatment for 24 h in AML12 cells (Scale bar, 50 μm) (Representative images are shown for each condition from one of three biologically independent experiments.). **d** Immunofluorescence of

PPARα with Importazole (IPZ, 10 μM), Leptomycin B (LMB, 10 ng/μl), and/or HDCA (100 μM) treatment in AML12 cells. AML12 cells were pretreated with IPZ for 1 h and then treated with HDCA for 24 h or treated with LMB and HDCA combination for 24 h (Scale bar, 25 μm). Difference between groups were determined by unpaired two-tailed Student's t test (Representative images are shown for each condition from one of three biologically independent experiments.). **e** Total, nuclear, and cytoplasmic protein levels of CRM1, RAN, and PPARα after HDCA treatment (100 μM) for 24 h in AML12 cells (n = 3). Difference between groups were determined by unpaired two-tailed Student's t test. **f** Total, nuclear, and cytoplasmic protein levels of CRM1 and RAN in the liver of mice treated as in Fig. 2a (n = 6 per group). Difference between groups were determined by one-way ANOVA test followed by Tukey's multiple comparison. The findings in **a–b** and **e** were confirmed in two independent experiments. Data are presented as mean values ± SEM. The average of gene and protein expression in control group is normalized as 100%. Source data are provided as a Source Data file.

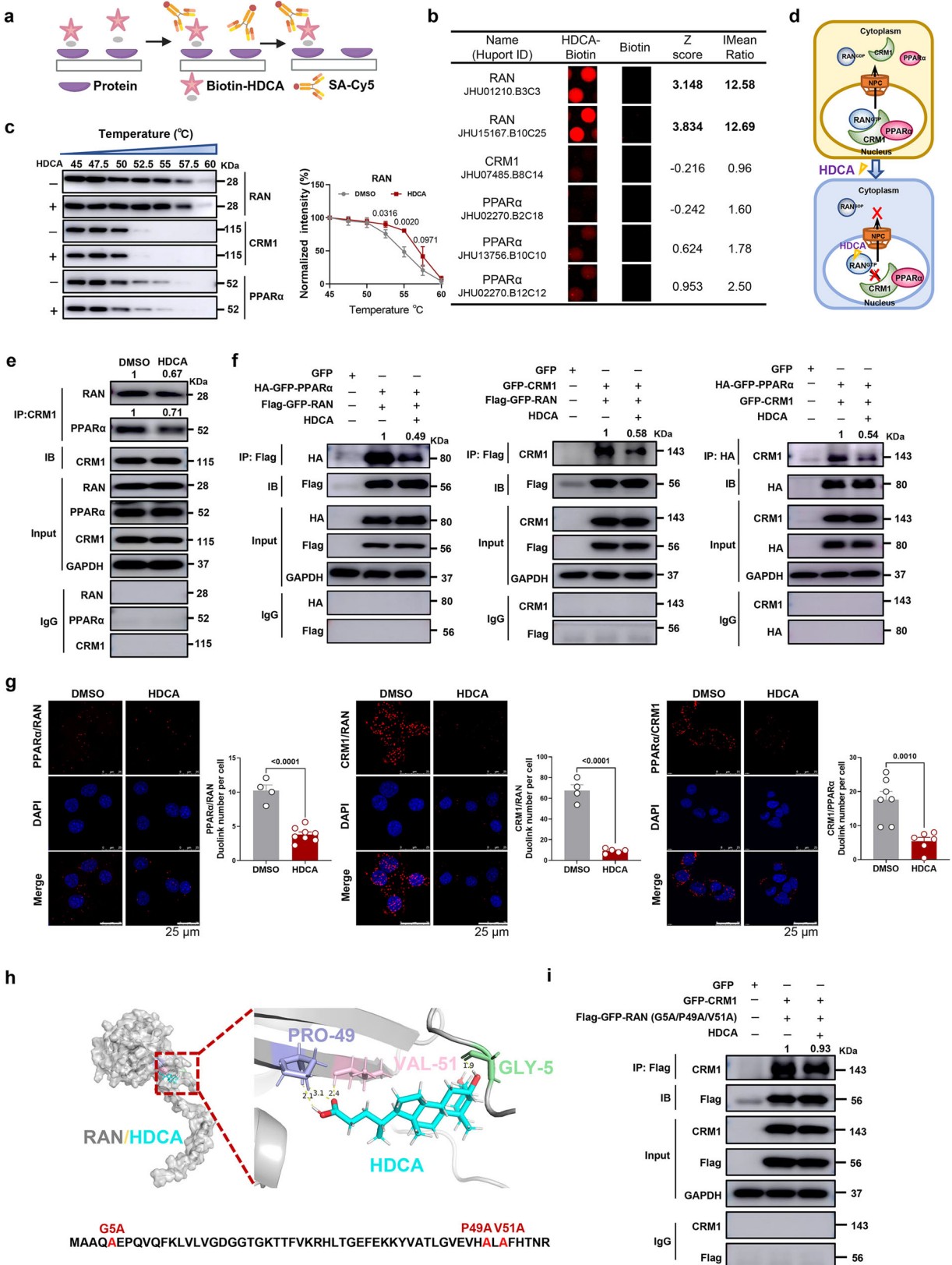

Some limitations in the current study include, first, whether HDCA can compete with GTP/GDP to bind to RAN was not investigated. Second, the exact molecular mechanism of how HDCA affects the interaction of RAN and CRM1 was not revealed. Further investigations on these aspects are warranted.

## Methods

### Clinical sample collection

Samples of NAFLD subjects were derived from a clinical study, which was led by Shuguang Hospital affiliated to Shanghai University of Traditional Chinese Medicine. The protocol was approved by

**Fig. 7 | HDCA directly binds with RAN to inhibit the formation of RAN/CRM1/
PPARα export heterotrimer. a** Schematic showing steps for identifying HDCA-
binding proteins using microarrays fabricated with recombinant human proteins.
Created with BioRender.com. **b** Magnified image of Bio-HDCA binding to RAN,
CRM1, PPARα spots on the protein array. Z score and I mean ratio are shown.
**c** Representative western blots showing thermostable individual proteins following
indicated heat shocks in the presence (+) or absence (−) of HDCA (100 μM) by
CETSA. Quantification of thermostable RAN from CETSA ($n = 3$ biologically inde-
pendent experiments). **d** Schematic showing interaction of HDCA with RAN/CRM1/
PPARα export complex. **e** Co-IP assay of the association of CRM1/RAN and CRM1/
PPARα in the presence of 100 μM HDCA in AML12 cells. **f** Co-IP assay of the asso-
ciation of Flag-GFP-RAN and HA-GFP-PPARα, Flag-GFP-RAN and GFP-CRM1, HA-
GFP-PPARα and GFP-CRM1 in the presence of 100 μM HDCA in HEK293T cells.

**g** Representative PLA images and duolink spot quantitation of PPARα/RAN, CRM1/
RAN and PPARα/CRM1 interactions in AML12 cells (Scale bar, 25 μm). Representa-
tive images are shown for each condition from one of 3 biologically independent
experiments. The graph shows the average number of PLA signals per cell, $n = 4–8$
random fields of view (42–60 cells of each condition). **h** Molecular docking
between HDCA and RAN (RAN protein (gray), HDCA (blue), GLY-5 (green), PRO-49
(purple) and VAL-51 (pink)). **i** Co-IP assay of the association of Flag-GFP-RAN (G5A/
P49A/V51A) and GFP-CRM1 in the presence of 100 μM HDCA (Gly-5, Pro-49, Val-51 of
RAN were all mutated to Ala). Experiments in **e**, **f**, and **i** were repeated indepen-
dently at least twice with similar results. Data are presented as mean values ± SEM.
Differences between groups were determined by unpaired two-tailed Student's *t*
test. Source data are provided as a Source Data file.

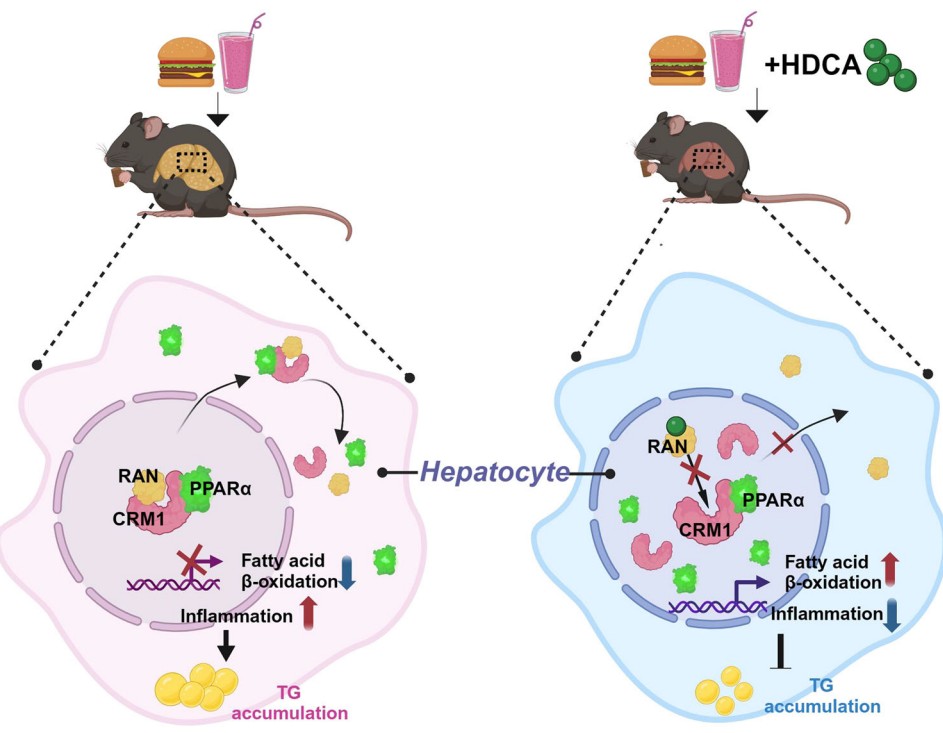

**Fig. 8 | A working model outlining the proposed mechanism.** Under HFHS diet
condition, HDCA intervention ameliorates NAFLD by inhibiting the formation of
nucleus-cytoplasm shuttling heterotrimer, RAN/CRM1/PPARα, through direct

binding with RAN protein, resulting in increased nuclear localization of PPARα and
activated fatty acid oxidation. Created with BioRender.com.

Institutional Review Board of Shuguang Hospital Affiliated to Shanghai
University of Chinese Medicine (No. 2017-548-31) and registered at the
Chinese Clinical Trial Registry (No. ChiCTR-IOR-17013491). The diag-
nostic criteria for NAFLD were referred to the Guidelines for the
Diagnosis and Treatment of Nonalcoholic Fatty Liver Disease (2010)
issued by Fatty liver and Alcoholic Liver Disease Group, Hepatology
Society of Chinese Medical Association. Healthy subjects were recrui-
ted from the Phase I clinical program of Good Clinical Practice Center
or the Physical Examination Center of Shanghai Shuguang Hospital,
which was conducted in accordance with the Declaration of Helsinki.
The protocol was approved by Institutional Review Board of Shuguang
Hospital Affiliated to Shanghai University of Chinese Medicine (No.
2019-662-17-01). Written informed consent was obtained from all
subjects. 34 NAFLD subjects (30 males and 4 females, ALT > 80, CAP
score > 300) and 24 healthy subjects (23 males and 1 female, ALT < 40,
CAP score < 235), aged 18-65 years old, were included in this study. The
baseline characteristics are detailed in Supplementary Table 1. The
inclusion and exclusion criteria for NAFLD patients and healthy sub-
jects are supplied in the supplementary material. Sex was not

considered in this study. The blood samples were drawn after fasting
12 h and serum samples were prepared and immediately frozen at
−80 °C.

### Animal studies
All animal experiments were conducted under the Guidelines for Animal
Experiment of Shanghai University of Traditional Chinese Medicine and
all animals received humane care according to the criteria outlined in
the Guide for the Care and Use of Laboratory Animals. Animal protocols
were approved by the Institutional Animal Ethics Committee of Shang-
hai University of Traditional Chinese Medicine (PZSHUTCM200417001,
PZSHUTCM200430002, and PZSHUTCM2306200001). Male C57BL/6
wild type mice and *Pparα*−/− mice were purchased from Vital River
Laboratory Animal Technology Co., Ltd. and Cyagen Biosciences (Suz-
hou) Inc., respectively. After 1 week acclimatization, all the mice were
maintained in specific-pathogen-free (SPF) environment under a 12 h
light/12 h dark cycle at 23°C–24°C with 60%±10% relative humidity.
Each animal was checked for its suitability according to animal welfare
authorities before individual experiment.

## Animal experiment 1: NAFLD mice model after 24-week HFHS diet feeding

Based on our preliminary observational experiment and other report[54], 24 weeks of HFHS feeding was sufficient for generating a typical NAFLD mouse model in C57BL/6 J mice. Thus, a total of 17 five-week-old male C57BL/6 J mice were randomly divided into two groups fed with a chow diet (C2018, SYSEBIO) and normal water or an HFD (60% fat, D12492, Research Diets) and supplemented with 30% sucrose in drinking water (HFHS group) for 24 weeks. The two groups were defined as Control (n = 8) and HFHS (n = 9) groups. Intraperitoneal glucose tolerance test (ipGTT) and intraperitoneal insulin tolerance test (ipITT) were administrated at the 22th and 23th week, respectively. The body weight and dietary intake were monitored during the feeding period.

## Animal experiment 2: the effect of HDCA in HFHS-fed C57BL/6 J mice

The dose for HDCA treatment was referred to previous reports that the effective concentration range of HDCA for suppressing atherosclerosis formation, reducing plasma cholesterol levels, and improving diabetes in rodent animals is from 0.05 to 1.25% supplied in diet[11,13–15,55,56]. A preliminary experiment was conducted to determine the dose effect of HDCA on ameliorating NAFLD by using three doses of 1.25%, 0.625%, and 0.3125% (H106315, Aladdin, 98% purity) supplied in the diet. Finally, medium-dose HDCA (0.625%) was chosen in the formal experiment. Additionally, based on our preliminary observational experiment, the feeding duration of 12 weeks was chosen because 12-week HFHS feeding is sufficient for inducing successful hepatic steatosis based on the steatosis score and NAS, which is more applicable for evaluating the anti-NAFLD effect of HDCA in mice compared to the long term 24-week feeding. Then, a total of 18 five-week-old male mice were randomly divided into two groups (control group n = 6, HFHS group n = 12). Control group and HFHS group were fed with chow diet or HFHS diet for 12 weeks (n = 6 per group). HDCA group were fed with HFHS diet for 4 weeks and then supplemented with 0.625% HDCA in the diet for another 8 weeks (n = 6). The body weight and dietary intake were monitored during the feeding period. ipGTT and ipITT were administrated at the 11th and 12th week, respectively.

## Animal experiment 3: the effect of HDCA in HFD-fed *ob/ob* mice

A total of 10 five-week-old *ob/ob* C57BL/6 J male mice were fed with an HFD diet with (n = 5) or without 1.25% HDCA (n = 5) for 8 weeks.

## Animal experiment 4: the effect of HDCA in HFHS-fed *Pparα*⁻/⁻ mice

The control groups of wild-type mice (8-week-old, male, C57BL/6 N) and *Pparα*⁻/⁻ mice (8-week-old, male, C57BL/6 N) were fed with HFHS for 12 weeks. HDCA groups of wild-type mice and *Pparα*⁻/⁻ mice (8-week-old, male) were fed with HFHS for 4 weeks and then supplemented with 0.625% HDCA in HFHS diet for another 8 weeks. The body weight and dietary intake were monitored during the feeding period.

## Animal experiment 5: the effect of HDCA in HFHS-fed *Pparα*ʰᵉᵖ⁻/⁻ mice

To obtain hepatocyte-specific *Pparα*-deficient or control mice, *Pparα*ᶠˡᵒˣ/ᶠˡᵒˣ mice (8-week-old, male, C57BL/6 J)[57] were injected with AAV2/8-TBG-Cre or control virus AAV2/8-TBG-ZsGreen (Hanbio technology, Shanghai) via the tail vein, respectively. The mice were given HFHS diet the same day they were injected with the virus. The control groups of *Pparα*ᶠˡᵒˣ/ᶠˡᵒˣ and *Pparα*ʰᵉᵖ⁻/⁻ mice were fed with HFHS for 12 weeks. HDCA groups of *Pparα*ᶠˡᵒˣ/ᶠˡᵒˣ and *Pparα*ʰᵉᵖ⁻/⁻ mice were fed with HFHS for 4 weeks and then supplemented with 0.625% HDCA in the diet for another 8 weeks. The body weight and dietary intake were monitored during the feeding period.

At the end of all the experiments, overnight fasted mice were euthanized by deeply anesthetized with pentobarbital sodium (100 mg/kg, i.p.) followed by cervical dislocation. Tissues and intestinal content samples were harvested and immediately frozen at −80 °C for further analysis.

## ipGTT and ipITT

For the GTT, glucose (1 g/kg body weight) was administered via intraperitoneal injection after overnight fasting. The glucose level of tail vein was measured at 0, 15, 30, 60, 90, 120 min after glucose load. For the ITT, insulin (0.75 U/kg body weight) was administered via intraperitoneal injection after 4 h fasting. The glucose level of the tail vein was measured at 0, 15, 30, 60, 90, 120 min after insulin load. Roche ACCU-CHEK Performa was applied for blood glucose data collection.

## Serum biochemical assay, hepatic and fecal lipids, and inflammatory cytokines assay

The serum levels of triglycerides, total cholesterol, ALT, and AST were measured according to the instructions of specific kits (Nanjing Jiancheng Bioengineering Institute, China). The level of fasting insulin was measured by Insulin ELISA Assay (Merck Millipore, EZRMI-13K, USA). The serum level of β-hydroxybutyrate was determined using β-hydroxybutyrate test kit (MAK041, Sigma) according to the manufacturer's protocol. Hepatic and fecal lipids were extracted according to the optimized Folch method[58]. TC, TG, NEFA level was measured with the instruction manual (Nanjing Jiancheng Bioengineering Institute, China). Hepatic TNF-α, IL-6, IL-1β were detected by Mlbio ELISA kits (M1002095, M1063132, M1002293) following the manufacture's protocol.

## Histopathological evaluation

Liver samples were fixed in 4% paraformaldehyde, embedded in paraffin and sectioned (3 μm thickness). The sections were subjected to hematoxylin-eosin staining (H&E) staining and Sirius Red using staining standard procedures. Images were acquired by ImageScope (Leica Biosystems Imaging, Inc., USA) or Digital slice scanner (3D Histech, Pannoramic MIDI, Hungary). Liver pathology was scored by pathologists according to NAFLD activity scoring system[59]. The scores comprised steatosis (0–3), lobular inflammation (0–3), and ballooning (0–2) scores. Steatosis were graded based on the percentage of involved hepatocytes, 0 (<5%), 1 (5–33%), 2 (34–66%), and 3 (>66%). Inflammation was graded based on the number of inflammatory foci per field (200×), 0 (no foci), 1 (<2 foci), 2 (2–4 foci), and 3 (>4 foci). Ballooning was scored as 0 (none), 1 (few balloon cells), 2 (many balloon cells).

## Oil Red O staining

The liver and ileum tissue fixed in 4% paraformaldehyde for 4 h were dehydrated for another 24 h and embedded in a frozen medium. The slices (8 μm) were stained with freshly prepared Oil Red O dye (D027, Nanjing Jiancheng Bioengineering) following the manufacturer's protocol. The red lipid droplets were imaged using Digital slice scanner (3D Histech, Pannoramic MIDI, Hungary).

## BODIPY gavage

The overnight fasted mice were gavaged with an olive-oil-infused bolus of fluorescent-labeled fatty acid BODIPY dye (2 μg/g BODIPY™ 500/510 C1, C12, Thermo Fisher, D3828) (10 μl/g mouse) and harvested 2 hours later. Jejunum sections were flash-frozen with OCT and then stained with DAPI. Slides were imaged using Digital slice scanner (3D Histech, Pannoramic MIDI, Hungary).

## TUNEL staining

Samples of liver or ileum were fixed in 4% paraformaldehyde, embedded in paraffin, and sectioned (3-μm thickness). The sections were subjected to TUNEL staining using staining standard procedures

(Roche, 1168475910). Images were acquired by Digital slice scanner (3D Histech, Pannoramic MIDI, Hungary).

## Quantitative analysis of BAs

BAs quantification was performed by Metabo-Profile Biotechnology (Shanghai) Co., LTD. All of the bile acids standards were synthesized by Metabo-Profile lab or obtained from Steraloids Inc. (Newport, RI, USA) and TRC Chemicals (Toronto, ON, Canada). For serum pretreatment, 20 μL sample serum was added to the 180 μL acetonitrile/methanol (v/v = 80:20) mixed solvent containing 10 μL internal standard into the 96-well plate. After shaking for 20 min at $200 \times g$, centrifugation was performed, and the supernatant was transferred to a microcentrifuge tube for freeze-drying. The dried samples were redissolved with 1:1 acetonitrile/methanol (v/v = 80:20) and ultrapure water, centrifuged at 4 °C for 20 min. The supernatant was transferred to a 96-well plate, and the injection volume was 5 μL. For tissue and intestinal contents: 10 mg of samples were weighted and homogenized with 180 μL acetonitrile/methanol (v/v = 80:20) mixed solvent containing 10 μL internal standard. After centrifugation for 20 min at $15,000 \times g$ at 4 °C, the supernatant was transferred to 96-well plate and freeze-dried. After drying, the powders of actual samples, standard samples, and quality control samples were redissolved with 1:1 acetonitrile/methanol (v/v = 80:20) and ultrapure water, centrifuged at $15,000 \times g$ at 4 °C for 20 min. The supernatant was transferred to a 96-well plate for UPLC/TQ-MS analysis with a volume of 5 μL. All the samples were run in a randomized order to minimize systematic analytical errors and pooled with quality control samples. The peak annotation and quantification were performed by MassLynx v4.1 and TargetLynx V4.1 (Waters Corp., Milford, MA, USA).

## RNA sequencing analysis

RNA sequencing analysis was performed according to our previous published methods[60]. Briefly, total RNA was extracted by TRIzol method from frozen liver tissue. The library was constructed and sequenced by Shanghai Majorbio Bio-pharm Technology Co., Ltd. TruSeq™ RNA Sample Preparation Kit (Illmina, San Diego, CA) was used to construct RNA libraries. The high-throughput sequencing was performed via Illumina HiSeq XTEN/NovaSeq 6000 squencer (2 × 150 bp read length). SeqPrep (https://github.com/jstjohn/SeqPrep) and Sickle (https://github.com/najoshi/Sickle) were applied to trim and quality. Then clean reads were separately aligned to reference genome with orientation mode using HISAT2 (http://ccb.jhu.edu/software/hisat2/index.shtml)[61] software. The mapped reads of each sample were assembled by StringTie (https://ccb.jhu.edu/software/stringtie/index.shtml?t=example) in a reference-based approach[62]. The expression of each gene was calculated according to Fragments Per Kilobases per Million reads[63]. RSEM (http://deweylab.biostat.wisc.edu/rsem/) was used to quantify gene abundances. The differential expression analysis was performed using the DESeq2[64]. DEGs with $|\log_2 FC| > 1$ and $P$ adjust value ≤0.05 were considered to be significantly different expressed genes. The data were analyzed on the online platform of Majorbio Cloud Platform (www.majorbio.com).

## Proteomics analysis

The quantitative proteomics analysis was performed at the Shanghai Institute of Materia Medica, Chinese Academy of Sciences[65]. Mouse liver tissues were lysed in SDT lysis buffer. The lysates were homogenized with sonication, denatured, and reduced at 95 °C for 5 min and then centrifuged at $12,000 \times g$ for 25 min. The supernatants were collected and further quantified by a Bradford assay. Peptides were prepared following the Filter Assisted Sample Preparation procedure.

Peptide labeling was conducted with TMT 10-plex reagents according to the manufacturer's protocol (Thermo Fisher Scientific). To increase the depth of protein identification, high-pH reverse-phase liquid chromatography was used for peptide fractionation. The peptides were separated using a Waters reversed phase XBridge BEH C18 column (150 × 2.1 mm, 3.5 μm) at a flow rate of 0.2 mL/min using Agilent 1200 HPLC systems.

The proteome analysis was performed on an Orbitrap Q-Exactive (Thermo Fisher Scientific) platform connected to an online nanoflow EASY nLC1200 HPLC system (Thermo Fisher Scientific). Peptides were loaded on a self-packed column (75 μm × 150 mm, 3 μm ReproSil-Pur C18 beads, Dr. Maisch GmbH, Ammerbuch, Germany) and separated with a 120 min gradient for each sample at a flow rate of 300 nL/min. A homemade column oven maintained the column temperature at 50 °C. A data-dependent acquisition MS method was used, in which one full scan (350–1700 m/z, $R = 120,000$ at 200 m/z) at a target of $3 \times 10^6$ ions was first performed, followed by top 20 data-dependent MS/MS scans with higher-energy collisional dissociation (HCD) at a resolution of 60,000 at 200 m/z. Other instrument parameters were set as follows: 32% normalized collision energy (NCE), $1 \times 10^5$ AGC target, 120 ms maximum injection time, 1.0 m/z isolation window.

Raw mass spectrometry data were processed using Maxquant 1.6.5.0 (Thermo Scientific) against the human Uniprot database, with a false-discovery rater (FDR) < 0.01 at the level of proteins and peptides. Further bioinformatics analysis was conducted in DAVID (https://david.ncifcrf.gov) for Kyoto Encyclopedia of Genes and Genomes (KEGG) analysis. The fold changes >1.2 with $p$ values < 0.05 were considered as a cutoff for differential proteins.

## Cell culture and treatment

AML12 (alpha mouse liver-12) hepatocytes were purchased from National Collection of Authenticated Cell Cultures (SCSP-550) and cultured in DMEM/F-12 medium (Meilunbio, PWL107, China) supplemented with 10% fetal bovine serum, 1% ITS (10 μg/ml insulin, 5.5 μg/ml transferrin, 5 ng/ml selenium), 40 ng/ml dexamethasone and 1% penicillin-streptomycin, at 37 °C, 5% $CO_2$. RAW264.7 (a mouse macrophage cell line) (ATCC, TIB-71) and HEK293T (ATCC, CRL3216) were cultured in DMEM (Meilunbio, PWL0212, China) containing 10% fetal bovine serum (FBS), 1% penicillin-streptomycin, at 37 °C, 5% $CO_2$. No commonly misidentified cell line was used in this study. All the cell lines were routinely tested negative for mycoplasma contamination.

## Protein isolation and western bolting

Liver tissues and cell sample lysates were prepared by RIPA buffer supplemented with 1 mM PMSF and phosphatase inhibitors. The protein concentration was measured by BCA Protein Quantification Kit (20201ES90, Yeasen Biotechnology, Shanghai, China). Specifically, the nuclear protein was acquired by a Nuclear and Cytoplasmic Protein Extraction kit (Beyotime Institute of Biotechnology, Shanghai, China). Proteins were separated by SDS-PAGE electrophoresis and transferred to PVDF membranes. Membranes were then probed with anti-PPAR alpha (Abcam, ab126285, ab215270, 1:1000 for WB), anti-CD36 (Abcam, ab133625, 1:1000 for WB), anti-FABP1 (Proteintech, 13626-1-AP, 1:1000 for WB), anti-CPT1A (D3B3) (Cell singaling, 12252 S, 1:1000 for WB), anti-CPT2 (Proteintech, 26555-1-AP, 1:1000 for WB), anti-HMGCS2 (Cell singaling, 20940 S, 1:1000 for WB), anti-NF-κB p65 (Cell singaling, 8242 S, 1:1000 for WB), anti-Phospho-NF-κB p65 (Ser536) (Cell singaling, 3033 S, 1:1000 for WB), anti-CRM1 (Santa Cruz Biotechnology, sc-74454, 1:500 for WB), anti-RAN (Santa Cruz Biotechnology, sc-271376, 1:500 for WB), anti-Lamin B1 (Abcam, ab229025, 1:1000 for WB), anti-GAPDH (Proteintech, 60004-1-Ig, 1:5000 for WB), anti-β-Actin (Cell singaling, 4970, 1:1000 for WB), anti-FLAG (Sigma, F1804, 1:1000 for WB), anti-HA (Proteintech, 51064-2-AP, 1:1000 for WB) antibodies followed by incubation with HRP-conjugated secondary antibodies (CST, 7076, 1:5000 for WB) (ABclonal, AS014, 1:5000 for WB). The signals were detected by chemiluminescence using the Amersham

Imager 600 (GE, USA). All the unprocessed scans of bands were supplied in Source Data file or Supplementary information.

## Real-time quantitative PCR

Total RNA was extracted with Trizol reagent (15596018, Thermo Fisher Scientific) from cells or tissues. cDNA was synthesized by the High Capacity cDNA Reverse Transcription Kit (KR118-02, TIANGEN). Then, quantitative PCR (qPCR) was carried out using the SYBR Green Master Mix (11201ES03, Yeasen Biotechnology, Shanghai, China) according to the manufacturer's instructions. The primers were synthesized by BioSune Biotechnology (Shanghai) Co., LTD, and primer sequences of the genes are listed in Supplementary Table 2.

## Immunofluorescence

The cells were fixed with 4% paraformaldehyde solution for 15 min and rinsed twice by PBS, then permeabilized with 0.5% Triton X-100 (ST797, Beyotime) at room temperature followed by PBS washing for twice. After washing, the cells were further incubated with 10% donkey serum (Jackson Immuno Research, USA) for 1 h to block non-specific antibody binding, then incubated with primary antibodies to PPARα (Abcam, 126285, 1:100) and subsequently incubated with ABflo® 594-conjugated Goat Anti-Rabbit IgG (H + L) (ABclonal, AS039, 1:100) for 1 h. After incubating with DAPI (P0131, Beyotime) for 10 min, images were directly captured using confocal microscopy (Leica, TCS SP8).

Samples of the liver were fixed in 4% paraformaldehyde, embedded in paraffin, and sectioned (3 μm thickness). The sections were subjected to immunohistochemistry staining using standard procedures. The samples were incubated with primary antibodies to PPARα (Abcam, 215270, 1:100) overnight and subsequently incubated with ABflo® 488-conjugated Goat Anti-Rabbit IgG (H + L) (ABclonal, AS053, 1:100) for 50 min. After incubating with DAPI for 10 min, images were acquired by fluorescent microscope (NIKON ECLIPSE C1, Japan).

## Luciferase reporter gene assays

After 24 h co-transfection with pCMV-Script-hFXR and PGL4-Shp-TK firefly luciferase plasmids[66], followed by continuous 6 h 1% FBS DMEM medium, the HEK293T cells were exposed to DMSO or different concentration of HDCA (20 μM and 100 μM) for 24 h to test FXR activity. Cell activity assays and luciferase assays were then performed by CellTiter-Blue Reagent (G8080, Promega) and Firefly-Glo Luciferase Reaction (MA0519, Meilunbio), respectively. Data were collected by SpectraMax M5e (Molecular Devices, USA).

## Human proteome microarray

The HuProt™ 20 K Human Proteome Microarrays (CDI Laboratories, Baltimore, MD) were used in this study. The experiments were performed by Wayen Biotechnologies (Shanghai, China) according to the following procedure. Briefly, proteome microarrays were blocked with blocking buffer (5% BSA in 0.1% Tween 20, PBST) for 1 h at room temperature with gentle agitation. Biotin-HDCA and biotin were diluted to 100 μM in PBST and incubated on proteome microarray at room temperature for 1 h (Biotin-HDCA was synthesized by Xi'an Ruixi Biological Technology Co., Ltd). After washing with PBST three times, microarray was incubated with 0.1% Cy5-Streptavidn solution for 20 min at room temperature in the dark, followed by three 5-min washes in PBST. The microarray was spun dry at $100 \times g$ for 2 min and scanned with GenePix™ 4000B (Axon Instruments, CA). GenePix™ Pro v6.0 was used for data analysis. The median fluorescence signal at each site (F635_Median) was divided by the background (B635_Median) for data analysis, that is, original signal strength ($I$) = F635_Median/B635_Median. The median and SD of I were calculated to obtain the corrected data Z Score (corrected signal strength) for each site. Proteins with a Z Score greater than 2.8 were those that bind to HDCA.

The mean value (I mean) represents the mean value of the original signal strength of each protein. I Mean Ratio is the ratio between the Biotin-HDCA group and Biotin group. To call the candidates, the cutoff was set as a $P$ value ≤ 0.05 and I mean ratio ≥ 1.4.

## Cellular thermal shift assay (CETSA)

AML12 cells were treated with DMSO or 100 μM HDCA for 1 h and washed with PBS. The cell pellets were resuspended in 1 mL PBS with protease inhibitor. Cells were divided into 100 μl aliquots and heated with a thermal gradient from 40 °C to 65 °C for 3 min. After freeze-thawing three times with liquid nitrogen, the supernatant was acquired by centrifugation at $2000 \times g$ for 10 min at 4 °C, followed by Western blotting.

## Co-immunoprecipitation (Co-IP)

All the GFP/Flag/HA-tagged plasmids (HA_PPARα_pcDNA3.1(+)-C-eGFP, FLAG_RAN_pcDNA3.1(+)-N-eGFP, CRM1_pcDNA3.1(+)-N-eGFP, G5A/P49A/ V51A-FLAG_RAN_pcDNA3.1(+)-N-eGFP) were produced and sequencing confirmed by GenScript (Nanjing, China). Co-IP was performed following the protocol of Co-immunoprecipitation Kit (10007D, invitrogen). The equilibrated beads were incubated with the Flag antibody (Sigma-Aldrich, F1804, 1:50 for IP), HA antibody (Proteintech, 51064-2-AP, 1:50 for IP), CRM1 antibody (Santa Cruz Biotechnology, sc-74454, 1:25 for IP) or Control IgG antibody (Rabbit Control IgG ABclonal, AC005, 1:100 for IP, Mouse Control IgG ABclonal, AC011, 1:100 for IP) at room temperature for 30 min and then incubated with the protein extracts at 4 °C overnight. The magnetic beads were collected by magnetic separator. All the non-specifically bounded proteins were removed by wash buffer. The bounded proteins were eluted from the beads with elution buffer for 10 min at 70 °C, then followed by Western blotting analysis.

## Duolink PLA

The AML12 cells were treated with 100 μM HDCA or DMSO for 24 h. Duolink PLA was performed according to the protocol of Duolink® PLA (DUO092101, Sigma-Aldrich). The primary antibodies are anti-PPARα (Proteintech, 15540-1-AP, 1:25), anti-CRM1 (Santa Cruz Biotechnolog, sc-74454, 1:10), anti-CRM1 (ABclonal, A19625, 1:10), anti-RAN (Santa Cruz Biotechnolog, sc-271376, 1:20).

## Molecular docking

Docking analysis was carried out by Swiss-Dock software from the Swiss Institute of Bioinformatics (http://www.swissdock.ch/)[67], with default parameters. The Protein structure of GTP-binding nuclear protein RAN (UniProt: P62827) was chosen as a target using the target selection tab in Swiss-Dock, and the HDCA (PubChem CID: 5283820) structure was uploaded using the ligand molecular selection tab in Swiss-Dock. PyMOL software (version: 2.5.2) was applied for analysis of the interaction of residues between RAN and HDCA.

## Statistical analysis

Except for RNA sequencing and proteomics, most of the plots were generated by GraphPad Prism 9.0 (GraphPad Software, San Diego, USA). The differential analysis was performed using the two-tailed Student's $t$ test, Kruskal–Wallis test, Mann–Whitney $U$, two-sided Fisher's exact test, or one-way ANOVA followed by Tukey's multiple comparison in the Graphpad Prism 9.0. Correlations between BA and phenotype were performed using SPSS 21.0 or Graphpad Prism 9.0. All data are expressed as mean ± standard error of mean (SEM) unless otherwise noted. Significance was set at $p < 0.05$.

## Reporting summary

Further information on research design is available in the Nature Portfolio Reporting Summary linked to this article.

## Data availability

The mass spectrometry proteomics data have been deposited to the ProteomeXchange Consortium[68] via the iProX partner repository[69,70] with the dataset identifier PXD044613. RNA sequencing data have been deposited in Genome Sequence Archive (GSA)[71,72] under the accession code CRA012267. Human proteome microarray data have been deposited in GEO database under the accession code GSE241065. The protein structure of RAN was obtained from the human Uniprot database (UniProt: P62827). Source data are provided with this paper.

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

## Acknowledgements

We sincerely acknowledge Professor Hu Zhou and his laboratory members (Shanghai Institute of Materia Medica Chinese Academy of Sciences) for their help in the proteomics analysis. This work was supported by the National Natural Science Foundation of China (U21A20413 to H.L., 82273624 to L.S.), Clinical Research Plan of SHDC (SHDC2020CR2049B to Y.H.), Natural Science Foundation of Shanghai (20ZR1453900 to L.S.), Shanghai Excellent Academic Leaders Program (21XD1403500 to H.L.), National Natural Science Foundation of China (82222071 to C.X.), SIMM-SHUTCM Traditional Chinese Medicine Innovation Joint Research Program (2022, E2G807H to C.X. and Y.H.) and Shanghai Municipal Science and Technology Major Project (C.X.).

## Author contributions

J.Z. and X.H. performed most experiments, and data analysis, and drafted the manuscript. X.G., Y. Hong, W.Z., J.Y., Yi. L., Ya. L., N.Z., Y.B., H.W., J.M., W.H., and Z.L. helped with animal experiments and data analysis. Q.L. and Y.Z. involved in the clinical sample collection. Y. Lyu was responsible for RNA sequence analysis. X.K. guided the PLA experiment. W.J. helped in the study design. C.X. guided the animal experiments and mechanistic study. Y. Hu designed the clinical trial. L.S. is involved in the study design, project management, and manuscript writing. H.L. supervised the project and revised the manuscript. H.L. was the lead contact for the study.

## Competing interests

The authors declared no competing interests.
