## [Peer Review File · Nature Communications]

Hyodeoxycholic acid ameliorates nonalcoholic fatty liver disease by inhibiting RAN-mediated PPAR α nucleus-cytoplasm shuttlingREVIEWER COMMENTS

Reviewer #1 (Remarks to the Author):

In this manuscript, Zhong et al present results that characterize the function of the hydoxychoolic acid (HDCA) on NAFLD protection and suggest that it regulates the nuclear translocation of PPAR α , a key nuclear receptor involved in fatty acid oxidation. Interestingly, they show that HDCA and, more largely, HCA species are decreased in patients and mouse with NAFLD while HDCA is specifically anti-correlated with NAFLD characteristics. To achieve their demonstration, the authors smartly combine data from human cohort, in vivo studies with omics technologies and in cellulo approaches. Overall, the study highlights interesting new findings on bile acid function on NAFLD protection. However, some important items should be addressed to reinforce their conclusion on HDCA molecular function on PPAR α nucleus-cytoplasm shuttling.

Major points:

- Fig.1. While GHCA decreases in plasma of NAFLD patients, no difference are observed in serum or liver of NAFLD mouse models. In line with that, while HCA species do not differ in NAFLD patients, they are significantly lower in HFHS-fed mouse serum and tissues. The authors should discuss those discrepancies between species.
- Fig.2 & Fig.S5. Compared to physiological condition, the authors show that HDCA plasma and liver content at the end of the HDCA nutritional intervention are increased by 200 and 50 folds, respectively. This supra-physiologic HDCA content could be explored/comment in term of consequences notably on entero-hepatic cycle and production of conjugated-derived molecules.
- Fig.S2 & Fig.2. Steatosis, inflammation and NAFLD activity scores should be added to properly characterize the mouse model and compared/justified to use of a “short term” HFHS protocol instead of long-term diet administration for HDCA supplementation.
- Fig.2 E-F. While intra-hepatic TG content is relatively high in mouse fed the HFHS/HDCA diet compared to HFHS, steatosis score is hugely decreased upon HDCA treatment. The authors should clearly comment how the establish this score which is a currently missing information.
- Fig.2. ALAT plasma activity is higher after a short term (12 weeks) vs long term HFHS diet

(24 weeks). This should be discussed as authors' interest in HDCA was coming out from this biomarker to characterize NAFLD patients and mouse model.

- Fig.2 & Fig.S6. In absence of inflammation scoring, authors should precise that HDCA prevent steatosis in HFD-fed ob/ob mouse. Moreover, histological characterization of fibrosis should be added to the phenotype analysis.
- Fig.3. As plasma ketone body quantification is a functionally reflect of PPAR α liver activity, it will be of interest to measure β -hydroxybutyrate in plasma of those mice.
- Fig.5. As PPAR α KO mice are overweight and spontaneously develop fatty liver when fed a chow diet, liver weight and liver ratio should be given. A quantification of TG content is necessary in this experiment. This will allow to compare this experiment with the experiment conducted in WT mouse (Fig.2). Of note, it is intriguingly that steatosis, inflammatory, and NAFLD activity scores reach the same number as in WT mice (Fig.2).
- Fig.6A & B. In cellulo experiments on FXR activity in HEK293T (A) are not convincing compared to agonist treatment in ALM12 cells (B). Please comment in figure legend which construct has been used for this assay.
- Fig.7. CoIP realized when both partners are co-overexpressed can not show specific interaction. To further characterize the molecular way of action of HDCA on PPAR α nucleo-cytoplasmic shuttling, this part of the work need to be reinforced to be fully convincing. Please, the plasmid and, especially, the promoter used in experiments, need to be given in Mat&Med section. PLA assay should also be performed to characterize PPAR α /CRM1 interaction w or w/o HDCA.

Minor Points:

- Fig.1. Clinical data, authors should mention sex ratio in healthy and NAFLD subjects.
- Fig.2. "Ballooning score was 0 in all mice". Authors should be conscious that hepatocyte ballooning is discussed and seems to be dependent on NAFLD mouse model or mouse genetic background used in the study. The NAFLD activity scoring method used in this study should be adapt in consequences and clearly explain in Mat & Med section.
- Fig.6B. To be more convincing, other PPAR α target genes should be tested.

Reviewer #2 (Remarks to the Author):

In this manuscript, author demonstrated that hyodeoxycholic acid (HDCA) feeding could decrease NAFLD development, and they believed that HDCA interfered the complex of RAN/CRM1/PPARalpha, which disrupted the nuclear-cytoplasmic trafficking of PPARalpha and its function of fatty acid oxidation. Several points need to be considered.

1. Increased lipid accumulation causes NAFLD and thus disrupts bile acid homeostasis, but not the other way around. Decreased HDCA in NAFLD patients should be the result but not the cause. In addition, HDCA represents a very small fraction of the total bile acid pool, and authors' data showed this as well (serum and liver HDCA are at nmol/L and nmol/g level). It's hard to believe that the clinical effects from BA change in NAFLD patients result from this minor but not the other major fraction of BA species (uM), such as DCA, CDCA, etc.

What is the HDCA level in mouse liver after HDCA feeding?

2. As a secondary bile acid, HDCA is one of the metabolic byproducts of intestinal bacteria. Authors' data also showed that majority of HDCA existed in the gut. The high level of HDCA might directly affect the lipid and cholesterol absorption in the gut, then reduce the lipid and cholesterol accumulation in the liver indirectly. Did authors check the lipid absorption in the gut, though HDCA feeding didn't affect the food/energy uptake? In addition, HDCA has been shown to activate TGR5/GLP1 signaling pathway in the intestine and contribute to the insulin resistance.

3. What was the rationality for the HDCA feeding dose? Although HDCA is a hydrophilic acid and its toxicity is lower than DCA, some studies have shown it can cause apoptosis after extended exposure.

4. Steatosis in NAFLD is typically centered on the central veins. Figure 1 panel D, HFHD group showed quite different from Figure 5E.

5. It's also weird to show all the relative gene/protein expression level in the control/vehicle group as 100.

Reviewer #3 (Remarks to the Author):

In this manuscript, Zhong et al. found that the serum HDCA significantly reduced in NAFLD patients, and HDCA showed strongly inverse correlation with clinical parameters of NAFLD. Meanwhile, It was verified that HDCA can improve diet-induced NAFLD by animal experiments. The mechanism by which HDCA ameliorated diet-induced NAFLD is that HDCA hindered the formation of RAN/CRM1/PPAR α shuttle heterotrimer by direct binding with RAN protein, thereby leading to the accumulation of nuclear PPAR α . This study uncover a novel mechanism underlying the anti-NAFLD effect of HDCA and provide the rationale for novel therapeutic strategy on NAFLD. Overall, the work is well planned and the findings are supported by the results and this manuscript is well organized with good logic. Therefore, it could be considered for publication after addressing the following issues.

1.The information of the recruited patients is not comprehensive. Are they taking drugs for the treatment of NAFLD? Such as hepatoprotectants, hypoglycemic and lipid-lowering drugs? and will these drugs affect the research results?

2.To verify that the anti-NAFLD effect of HDCA is PPAR α -dependent, it is recommended that the anti-NAFLD effect of HDCA should be tested by liver-specific Ppar α knockout mice in animal experiments.

3.In animal experiments 1-4 (SUPPLEMENTARY METHODS AND MATERIALS), Why are the modeling time and dose of HDCA inconsistent? Is there any literature support for experimental scheme? Please add references. In animal experiment 3, The grouping and experimental scheme of HDCA group are not clearly described (line 47).

4.Some notes (e.g. Figure S1, and S6) are not corresponding to the right figures, which should be double-checked and revised.

5.Several images on KEGG enrichment analysis and heatmap of proteins related to fatty acid metabolism are blurry. It is suggested to improve the resolution of pictures in the article.

Reviewer #1 (Remarks to the Author)

In this manuscript, Zhong et al present results that characterize the function of the hydoxycholeic acid (HDCA) on NAFLD protection and suggest that it regulates the nuclear translocation of PPAR α , a key nuclear receptor involved in fatty acid oxidation. Interestingly, they show that HDCA and, more largely, HCA species are decreased in patients and mouse with NAFLD while HDCA is specifically anti-correlated with NAFLD characteristics. To achieve their demonstration, the authors smartly combine data from human cohort, in vivo studies with omics technologies and in cellular approaches. Overall, the study highlights interesting new findings on bile acid function on NAFLD protection. However, some important items should be addressed to reinforce their conclusion on HDCA molecular function on PPAR α nucleus-cytoplasm shuttling.

Major points:

1. Fig.1. While GHCA decreases in plasma of NAFLD patients, no difference are observed in serum or liver of NAFLD mouse models. In line with that, while HCA species do not differ in NAFLD patients, they are significantly lower in HFHS-fed mouse serum and tissues. The authors should discuss those discrepancies between species.

Response: This question is an important issue in biological study where discrepant data between species are commonly observed. In our current study, decreased GHCA was observed in plasma of NAFLD patients compared to healthy controls, while level of GHCA in liver was not different between groups, and was not detected in serum of mice. We think the reason for this discrepancy is possibly due to the fact that the majority of conjugated-bile acid in rodents is taurine-conjugated, rather than glycine-conjugated bile acids. So the GHCA content in serum of mice might be too low to reach the detection limit (1 nM) of bile acid based on our method.

As for HCA species, the HCA species in humans mainly include HDCA, HCA, GHCA, and THCA, with the proportions of THDCA and THCA being very low, while the HCA species in rodents mainly include HDCA, HCA, THCA, and THDCA, with the proportion of GHCA and THCA being very low. Our results showed in the serum of NAFLD patients, the levels of HCA species, HDCA, and GHCA were lower than health controls (Fig. 1c-h), while HCA species, HDCA, and THDCA were significantly lower in HFHS-fed mice than control group (Fig. 1k-m). We speculate that these discrepancies between species might be due to the differences in the formation of primary bile acids in the liver, and transportation of bile acids in the liver and intestine, re-absorption of bile acids in the intestine, as well as the composition and function of gut microbiota.

In summary, these discrepancies of bile acid changes between species might be due to the differences in formation, transportation, re-absorption, as well as gut microbiota-mediated production of secondary bile acids. The related information has been added to the discussion section in the revised text (Page 13, Line 319-325).

2. Fig.2 & Fig.S5. Compared to physiological condition, the authors show that HDCA plasma and liver content at the end of the HDCA nutritional intervention are increased by 200 and 50 folds, respectively. This supra-physiologic HDCA content

could be explored/comment in term of consequences notably on entero-hepatic cycle and production of conjugated-derived molecules.

Response: We would like to express our sincere gratitude to the reviewer for this valuable comment. After 8 weeks of HDCA intervention, we detected the content of bile acids in the serum, liver, and feces of mice, and found that the bile acid profiles in these three parts significantly changed (**new Fig. S5**). In particular, HDCA and its conjugated forms (THDCA and GHDCA) were the highest elevated bile acids. HDCA and THDCA consistently increased in serum and liver of HDCA supplemented mice, and reached relative high concentrations. It is possible the unconjugated HDCA was transformed to tauro-conjugated THDCA in mice. Since we have found that HDCA intervention ameliorated NAFLD in a PPAR α -dependent way, and also confirmed that HDCA could directly regulate PPAR α nuclear localization *in vitro*, we then asked whether the observed effect of HDCA was partly contributed by the increased THDCA, which was elevated in response to HDCA intervention. Our *in vitro* data indicated that THDCA did not affect PPAR α subcellular localization (**new Fig. S13**), suggesting the anti-NAFLD effect of HDCA through PPAR α mainly rely on HDCA, but not its conjugated form.

These results have been added in **New Fig. S5** (The bile acid profiles after HDCA intervention) and **New Fig. S13** (The effect of THDCA on PPAR α subcellular localization) in the revised manuscript, and the related information has been added to the **Result** (Page 6, Line118-119; Page 8, Line199-201) and **Discussion** sections (Page 14, Line 344-348).

3. Fig.S2 & Fig.2. Steatosis, inflammation and NAFLD activity scores should be added to properly characterize the mouse model and compared/justified to use of a “short term” HFHS protocol instead of long-term diet administration for HDCA supplementation.

Response: We have incorporated Steatosis, inflammation and NAFLD activity scores in **new Fig.S2 g**.

As for the two time points (24 or 12 weeks) of HFHS protocol used in our current study, we first conducted an observational experiment by feeding C57 mice with HFHS for 4, 8, 12 and 24 weeks, respectively, in which the extent of hepatic steatosis was evaluated based on liver histology with H&E staining. We found that 8-week HFHS feeding resulted in the presence of hepatic steatosis, and the extent of hepatic steatosis progressed time-dependently. Obviously, fatty liver was well-established in mice by 24-week HFHS feeding (**Response to reviewer Fig. 1**), which is consistent with other reports (PMID: 29222421, PMID: 27261415). Based on these observations, a long term 24-week HFHS feeding protocol was first used for generating a typical NAFLD mouse model in C57 mice, and samples from these mice were subjected for identifying NAFLD-related bile acids. In the subsequent HDCA intervention experiments, we chose 12 weeks, instead of 24 weeks, is due to the observation of HFHS diet impact on hepatic steatosis formation from 4 to 24 weeks. It is obvious that 12 weeks HFHS feeding is sufficient for inducing hepatic steatosis, based on the steatosis score and NAS (**Fig. 2d-e**). As a result, it is more applicable for evaluating the anti-NAFLD effect of HDCA in mice based on 12 weeks HFHS feeding compared to the long term 24 weeks. We observed that continued 8 weeks of HDCA supplementation was sufficient for ameliorating NAFLD in 12-week HFHS fed wild type

mice. These information have been added to the **revised Methods section** (Page 18, Line 443-447).

Response to reviewer Figure 1: Representative images of liver H&E staining after 4, 8, 12, 24 weeks high fat high sugar (HFHS) diet intervention. (Scale bar, 50 μ m). $n=8-10$ per group.

4. Fig.2 E-F. While intra-hepatic TG content is relatively high in mouse fed the HFHS/HDCA diet compared to HFHS, steatosis score is hugely decreased upon HDCA treatment. The authors should clearly comment how the establish this score which is a currently missing information.

Response: The scoring criteria for the liver steatosis score were primarily based on a publication in Hepatology in 2005 (PMID: 15915461). The Pathology Committee of the NASH Clinical Research Network proposed Non-alcoholic Steatohepatitis Clinical Research Network (NASH CRN) scoring system after validation in clinical, which has been widely used in research studies. In this scoring system, the evaluation of hepatic steatosis is semi-quantitative, with a score of 0 for less than 5% steatosis, a score of 1 for 5%-33% steatosis, a score of 2 for >33%-66% steatosis, and a score of 3 for >66% steatosis. We have supplemented the description on the scoring criteria in the **revised Methods section** (Page 20, Line 500-506).

As for the results of hepatic TG and steatosis score value, it should be noted that the steatosis score is semi-quantitative, while liver TG is quantitative. Additionally, the liver steatosis may be affected by the location and angle of tissue sampling, as the presence of hepatic steatosis may vary in different areas of liver. The hepatic TG content in control group was around 0.034 mmol/g liver (mean). HFHS feeding increased the hepatic TG to 0.075 mmol/g (mean), which meant the difference of liver TG between control and HFHS groups was 0.041 mmol/g (0.075-0.034). The hepatic TG level of HDCA treatment group was 0.058 mmol/g liver (mean), which meant the difference of liver TG between HFHS and HDCA groups was 0.017 mmol/g (0.075-0.058). Therefore, the HDCA treatment reduced about 41% of HFHS-induce liver TG accumulation ($((0.017/0.041)*100%=41\%$, **response to reviewer Fig. 2**). For steatosis score, the control group shows score of 0, and the average score of HDCA group is about 1.5, which is half to HFHS group (**Fig. 2e**), so the

changes of liver TG by HDCA was almost consistent with steatosis score in our study.

Response to reviewer Figure 2. Liver TG in control, HFHS or HDCA groups (Figure 2f in the manuscript). The C57BL/6 mice in control group and HFHS group were fed with chow or HFHS diet for 12 weeks. HDCA group were fed with HFHS for 4 weeks and then supplemented with 0.625% HDCA in the HFHS diet for another 8 weeks ($n=6$ per group). Values are means of each group.

5. Fig.2. ALT plasma activity is higher after a short term (12 weeks) vs long term HFHS diet (24 weeks). This should be discussed as authors' interest in HDCA was coming out from this biomarker to characterize NAFLD patients and mouse model.

Response: Thanks for this important suggestion. During our original submission, the ALT activity of mice after 12 or 24 weeks HFHS feeding were measured separately with two batches of ALT activity assay kits, in which the ALT values were calculated based on two different calibration curves. In this revision, we re-measured the serum ALT levels of both batches of mice simultaneously using same kit. The results showed that the serum ALT level of mice after 12-weeks HFHS feeding was approximately 57.58 U/L (**new Fig. 2f**), while it was approximately 79.43 U/L after 24-weeks HFHS feeding (**new Fig. S2d**). The data were renewed in the revised manuscript (**new Fig. 2f and S2d**).

6. Fig.2 & Fig.S6. In absence of inflammation scoring, authors should precise that HDCA prevent steatosis in HFD-fed *ob/ob* mouse. Moreover, histological characterization of fibrosis should be added to the phenotype analysis.

Response: Thanks. We have revised the description according to suggestion (Page 6, Line 137-139, 141-142). In addition, we have added the NAS scores and Sirius Red staining images of *ob/ob* mouse liver tissue in **new Fig. S7c**, as well as the Sirius Red staining images of C57 mouse liver tissue after HDCA intervention in **new Fig. S6a**. The fibrosis scores of these mice were presented in the table below (**Response to reviewer Table 1 and 2**). The scoring criteria for fibrosis score was based on a publication in Hepatology in 2005 (PMID: 15915461), in which fibrosis was scored as stage 0 (no fibrosis), stage 1a (mild perisinusoidal fibrosis), stage 1b (moderate perisinusoidal fibrosis), stage 1c (portal/periportal fibrosis), stage 2 (perisinusoidal and portal/periportal fibrosis), stage 3 (bridging fibrosis), and stage 4 (cirrhosis). Our results indicated that liver fibrosis was not significant in either *ob/ob* mice or HFHS-fed wild type mice, except for some mild fibrosis.

Response to reviewer Table 1. The fibrosis score of HFD-fed *ob/ob* mice

Group	Mouse 1	Mouse 2	Mouse 3	Mouse 4	Mouse 5
HFD group	1a	1a	1a	0	0
HDCA group	1a	1b	0	1b	1b

Response to reviewer Table 2. The fibrosis score of HFHS-fed wild type mice

Group	Mouse 1	Mouse 2	Mouse 3	Mouse 4	Mouse 5	Mouse 6
Control	0	0	0	0	0	0
HFHS	1b	1b	1b	1b	1a	2
HDCA	1b	1a	0	0	2	1b

7. Fig.3. As plasma ketone body quantification is a functionally reflect of PPAR α liver activity, it will be of interest to measure β -hydroxybutyrate in plasma of those mice.

Response: Thanks for the reviewer's suggestion. The levels of β -hydroxybutyrate in the serum of mice among groups were measured, and shown in the **new Fig. S11**. HDCA supplementation did increase the level of β -hydroxybutyrate compared to HFHS mice, which also showed higher level of β -hydroxybutyrate than that of control mice. It should be noted that increased β -hydroxybutyrate by high calorie diet (HFD, HFHS) feeding were previously reported (PMID: 29925686, 20233938, 22493093).

In addition, we also measured the expression of the ketogenic rate-limiting enzyme HMGCS2 and found that its expression was not changed after 12-week HFHS feeding, but significantly increased after HDCA intervention (**Fig. 4c**). The expression of HMGCS2 is strictly controlled by transcription factors such as PPAR α , CREB, SP1, COUP-TF, FKHL1, Foxa2, and HNF4 (PMID: 27983603, 10051425, 9291136, 12027802, 15616563, 9464279) and various post-translational mechanisms (PMID: 20203611, 2867762). In this study, the increase of nuclear PPAR α was consistent with the increase of HMGCS2 expression, suggesting that the enhancement of the ketogenesis by HDCA intervention is closely related to the activity of PPAR α . However, the upregulation of the ketogenesis in the HFHS group of mice may be related post-translational modification of HMGCS2 or other transcription factors besides PPAR α , and might be a compensatory manifestation of the body. The level of serum β -hydroxybutyrate has been included in **new Fig. S11**, and revised result and discussion sections in Page 8, Line 181-183; Page14, Line 339-344.

8. Fig.5. As PPAR α KO mice are overweight and spontaneously develop fatty liver when fed a chow diet, liver weight and liver ratio should be given. A quantification of TG content is necessary in this experiment. This will allow to compare this experiment with the experiment conducted in WT mouse (Fig.2). Of note, it is intriguingly that steatosis, inflammatory, and NAFLD activity scores reach the same number as in WT mice (Fig.2).

Response: Firstly, the successful construction of whole body PPAR α KO mice (*Ppara*^{-/-}) were verified by qPCR and Western blot (**new Fig. S14 a-b**). Then, to confirm their phenotype, we compared the liver weight and liver ratio of whole body PPAR α KO mice (*Ppara*^{-/-}) with wild type mice. The results showed that whole body PPAR α KO mice (*Ppara*^{-/-}) had elevated liver weight and liver ratio than WT mice (**New Fig. S14c**). In addition, the liver TG of *Ppara*^{-/-} mice has been included in the revised figure (**new Fig. 5e**). Based on liver TG and histology, our results suggested that deletion of whole body PPAR α abolished

the anti-NAFLD effect of HDCA, which is different with the observation in wild type mice.

Based on the NAS scoring system (PMID: 15915461), a score of 3 is for steatosis involved >66%. We found the steatosis score for most 12-week HFHS fed wild type mice already reached 3, which is the highest score for steatosis score. Thus, under HFHS-fed condition, most mice from wild type and *Ppara*^{-/-} groups showed similar steatosis score. Additionally, the inflammation score was higher *Ppara*^{-/-} mice than wild type mice, resulting to elevated NAS score in *Ppara*^{-/-} group. Together, our data clearly showed that deletion of whole body PPAR α abolished the anti-NAFLD effect of HDCA. These information have been added to **new Fig. 5** and **new Fig. S14** and result section (Page 9, Line 207-213).

9. Fig.6A & B. In cellulo experiments on FXR activity in HEK293T (A) are not convincing compared to agonist treatment in AML12 cells (B). Please comment in figure legend which construct has been used for this assay.

Response: HEK293T cell line is a widely used cell model for investigating functions of exogenous genes because of its characters of rarely expressed endogenous receptors and easiness for transfection. Since the promoter region of *shp* gene contains the FXR binding site FXRE, a classical downstream target gene of FXR, we then simultaneously transfected HEK293T cells with the PGL4-Shp-TK firefly luciferase plasmid containing FXRE and the human FXR expression plasmid (pCMV-Script-hFXR) (PMID: 30397356) to evaluate whether HDCA could activate FXR directly. The results showed that HDCA was a weak agonist of FXR in HEK293T cells (**Fig. 6a**). Nevertheless, compared to HEK293T cells, AML12 cell line is much more complicated where the increased expression of *shp* gene in the context of HDCA was probably due to either direct activation of FXR or other undetermined factors induced under HDCA intervention. So it is reasonable that stronger activation on FXR by HDCA in AML12 than HEK293T cells, as shown in Fig. 6b (Page 9 Line 231- Page 10 Line 236). In addition, the plasmid information has been added to the Method section and Figure legend (Page25, Line 644-645; Page 37, Line 1004-1009).

10. Fig.7. CoIP realized when both partners are co-overexpressed can not show specific interaction. To further characterize the molecular way of action of HDCA on PPAR α nucleo-cytoplasmic shuttling, this part of the work need to be reinforced to be fully convincing. Please, the plasmid and, especially, the promoter used in experiments, need to be given in Mat&Med section. PLA assay should also be performed to characterize PPAR α /CRM1 interaction w or w/o HCDA.

Response: We fully agree with this important comment. Previously, we were unable to obtain satisfactory results in AML12 cells when detecting endogenous association of CRM1/RAN and CRM1/PPAR α in the presence of HDCA, because the size of PPAR α and RAN are 52 kDa and 28 kDa respectively, which are similar to the heavy chain (50 kDa) and the light chain (25 kDa) of IgG. During revision, we optimized the experimental protocol by adjusting the pH of the elution buffer and found HDCA significantly weakened the interaction between CRM1 and RAN, as well as between CRM1 and PPAR α , confirming its interfering effect on the formation of RAN/CRM1/PPAR α complex at an endogenous level (**new Fig. 7e**). Additionally, we also performed PLA assay to detect CRM1/PPAR α interaction. PLA data revealed that the endogenous interaction of PPAR α /RAN,

RAN/CRM1, and CRM1/PPAR α were significantly disrupted in the presence of HDCA (**new Fig. 7g**). These new data further supported our hypothesis, and have been added to **new Fig. 7 e and g** and the result section (Page 11, Line 273-286). The information of plasmids, including the promoter used in experiments, have been added in Methods section (Page 26, Line 680-683).

Minor Points:

11. Fig.1. Clinical data, authors should mention sex ratio in healthy and NAFLD subjects.

Response: Thanks for pointing out this issue. We have supplemented the information of sex ratio of clinical samples in **new Table S1**. In detail, there were one female among the 24 healthy subjects, and four females in the 34 patients with NAFLD. There was no significant difference in sex ratio between the two groups ($P > 0.05$). Moreover, we did not find any sex-related difference in profile of serum bile acids by using OPLSDA analysis (**Response to reviewer Fig. 3**).

Response to reviewer Figure 3. OPLSDA plot of serum BA profile in health ($n=24$) and NAFLD ($n=34$) individuals. Sex is labelled.

12. Fig.2. “Ballooning score was 0 in all mice”. Authors should be conscious that hepatocyte ballooning is discussed and seems to be dependent on NAFLD mouse model or mouse genetic background used in the study. The NAFLD activity scoring method used in this study should be adapt in consequences and clearly explain in Mat & Med section.

Response: We appreciate the reviewer’s comments. Consistent with previous report that ballooning was rarely found in the experimental NAFLD models (PMID: 25535951), we did not find typical ballooning hepatocytes in our current liver samples. It may be due to the modeling method and duration, as well as the genetic background of mice (Page 6, Line 122-124).

In this study, the NAFLD activity scoring method was specifically according to Non-alcoholic Steatohepatitis Clinical Research Network (NASH CRN) scoring system proposed by Pathology Committee of the NASH Clinical Research Network published in *Hepatology* in 2005 (PMID: 15915461). The scores comprised steatosis (0-3), lobular inflammation (0-3), and ballooning (0-2). Steatosis were graded based on the percentage of involved hepatocytes, 0 (<5%), 1 (5–33%), 2 (34-66%) and 3 (>66%). Inflammation was graded based on the number of inflammatory foci per field (200 \times), 0 (no foci), 1 (<2 foci),

2 (2-4 foci) and 3 (>4 foci). Ballooning was scored as 0 (none), 1 (few balloon cells), 2 (many balloon cells). This scoring method has been supplemented in the **revised Methods section** (Page 20, Line 500-506).

13. Fig.6B. To be more convincing, other PPAR α target genes should be tested.

Response: We have added the expression of other PPAR α target genes in the **new figure 6B** including *Cpt1*, *Acadl*, *Acadm*, and *Acadvl*, and the results are consistent with our original conclusion.

Reviewer #2 (Remarks to the Author):

In this manuscript, author demonstrated that hyodeoxycholic acid (HDCA) feeding could decrease NAFLD development, and they believed that HDCA interfered the complex of RAN/CRM1/PPAR α , which disrupted the nuclear-cytoplasmic trafficking of PPAR α and its function of fatty acid oxidation. Several points need to be considered.

1. Increased lipid accumulation causes NAFLD and thus disrupts bile acid homeostasis, but not the other way around. Decreased HDCA in NAFLD patients should be the result but not the cause. In addition, HDCA represents a very small fraction of the total bile acid pool, and authors' data showed this as well (serum and liver HDCA are at nmol/L and nmol/g level). It's hard to believe that the clinical effects from BA change in NAFLD patients result from this minor but not the other major fraction of BA species (μ M), such as DCA, CDCA, etc. What is the HDCA level in mouse liver after HDCA feeding?

Response: It is definitely correct that excessive lipid accumulation causes the development of NAFLD directly. However, there are lots of factors that determining the homeostasis of lipid metabolism in human body such as amount of dietary fat intake, lipid metabolism, gut microbiome and genetics as well. Taken bile acids into consideration, the disrupted bile acid profile in disease like NAFLD, on one hand, might be consequence of NAFLD; on the other hand, disrupted bile acids could also contribute to lipid accumulation by affecting the multiple steps of lipid metabolism which has been well demonstrated in various studies (PMID: 28214524, PMID: 33636214). Nevertheless, the exact role of some specific bile acid in NAFLD development is highly valued and widely investigated, but is not fully understood.

It is a very important issue of the reviewer's concern on the relatively low concentration of HDCA compared to other bile acids with high content. Actually, there are numerous studies on the biological functions of those bile acids with high abundance such as DCA, CDCA, TCA etc in diseases like gastric intestinal metaplasia, acute myeloid leukemia, ulcerative colitis (PMID: 36067404, 36084349, 27261622). However, this does not mean the bile acid with low quantity is definitely "meaningless" for diseases. Previously, studies have reported that HDCA suppresses atherosclerosis, reduces cholesterol, and improves blood glucose homeostasis (PMID: 23752203, 11483626, 33338411, 25189147, 7619860, 35697422). In our current study, we screened and identified HDCA was one characterized

bile acid with consistent and low content in NAFLD patients and mice compared with their healthy controls. Based on this observation, we then hypothesized that additional HDCA supplementation might exert benefits against NAFLD development because there is no report on the role of HDCA on NAFLD, and HDCA is also an approved hypolipidemic drug in clinic. In other words, we actually did not rule out the possibility that other disrupted bile acids either with high or low abundance might have certain roles in NAFLD development. Based on our results, we concluded that dietary HDCA supplementation attenuated NAFLD development by activating PPAR α -regulated fatty acid oxidation pathway, leading to reduced lipid accumulation in hepatocytes, which is consistent with the concerns raised by reviewer.

In addition, we detected the bile acid profile in the liver, serum, and feces of HDCA-treated mice (**new Fig. S5**), the HDCA concentration in the liver increased to 65-242 nmol/g. Please also refer to the response to the Question 2 of Reviewer 1. We revised the statements on the role of HDCA in NAFLD development throughout the manuscript.

2. As a secondary bile acid, HDCA is one of the metabolic byproducts of intestinal bacteria. Authors' data also showed that majority of HDCA existed in the gut. The high level of HDCA might directly affect the lipid and cholesterol absorption in the gut, then reduce the lipid and cholesterol accumulation in the liver indirectly. Did authors check the lipid absorption in the gut, though HDCA feeding didn't affect the food/energy uptake? In addition, HDCA has been shown to activate TGR5/GLP1 signaling pathway in the intestine and contribute to the insulin resistance.

Response: In order to investigate whether HDCA affects lipid absorption in the intestine, we first extracted fecal lipids from 12-week HFHS-fed mice and HDCA-treated mice. Our results showed that HDCA supplementation significantly decreased intestinal cholesterol absorption because the fecal TC was elevated in HDCA group (**new Fig. S6d**), which is consistent with previous findings (PMID: 23752203, 7619860, 11483626). However, HDCA supplementation didn't affect the absorption of TG or non-esterified fatty acids (NEFA) as there was no difference of fecal TG and NEFA between HFHS and HDCA groups (**new Fig. S6d**). Next, to determine if enterocytes affect the efficiency of lipid absorption in response to HDCA, we analyzed enterocytes in the jejunum in two ways: BODIPY and Oil Red O staining. First, mice fed with chow diet, 12-week HFHS diet, or fed with HFHS for 4 weeks and then supplied with HDCA in HFHS for another 8 weeks (mice from Figure 2) were gavaged with an olive-oil-infused bolus of fluorescent-labeled fatty acid BODIPY dye. We then isolated the jejunum after 2 h and quantified lipid uptake facilitated by enterocytes. When compared with mice fed with chow diet, mice on HFHS diet displayed a significantly higher rate of lipid uptake, and there was no significant difference between HFHS and HDCA groups (**new Fig. S6e**). At the same time, we also used Oil Red O staining to detect the steady-state presence of lipids in enterocytes. Different from the observation in control group, in HFHS-fed mice, lipids accumulated along the whole villi, and HDCA intervention had the similar effect as HFHS group (**new Fig. S6e**). Since triglycerides are the majority lipids in hepatic steatosis (PMID: 29357123), we conclude the anti-NAFLD effect of HDCA might not due to the less lipid absorption in the gut.

As the reviewer mentioned, HDCA has been proven to improve insulin resistance by activating the TGR5/GLP1 signaling pathway in the intestine (PMID:33338411, 35697422), while the effects and targets of HDCA on the liver are still not completely clear. In this case, our study found that the regulatory effect of HDCA on PPAR α just fills this gap. The **new Fig. S6d-e**, as well as related description has been added to the result section (Page 6, Line 130-136; Page 14, Line 331-334).

3. What was the rationality for the HDCA feeding dose? Although HDCA is a hydrophilic acid and its toxicity is lower than DCA, some studies have shown it can cause apoptosis after extended exposure.

Response: The dose is vital for HDCA supplementation. In current study, the determination on the dose of HDCA was first referred to previous reports where HDCA was applied for improving atherosclerosis, cholesterol and diabetes in rodent animals at the dose from 0.05 to 1.25% supplied in diet (PMID: 23752203, 11483626, 33338411, 25189147, 7619860, 35697422). We then conducted a preliminary experiment to determine the dose effect of HDCA on ameliorating NAFLD by using three doses of 1.25%, 0.625%, and 0.375% in diet. Except for the only improvement in inflammation at 0.375% dose, obvious effects on improvement in both hepatic lipid accumulation and inflammation were observed at both 0.625% and 1.25% doses (**new Fig. S4**). So we finally chose 0.625% for the subsequent experiments. We have included these information in the **revised Methods section** (Page 18, Line 436-442).

As for whether HDCA will induce apoptosis or not, we tested the apoptosis status by using TUNEL fluorescence staining of liver and ileum tissues from mice that were fed with HFHS for 8 weeks with or without 0.625% HDCA supplementation in diet. Our results showed no apoptosis present among groups during the 8 weeks intervention (**new Fig. S6 b and c**). We have included these data in the **result section** (Page 6, Line 128-130). Whether more extended exposure of HDCA would result in apoptosis needs further evaluation.

4. Steatosis in NAFLD is typically centered on the central veins. Figure 2 panel D, HFHD group showed quite different from Figure 5E.

Response: Thanks for pointing out this issue. We re-analyzed all the H&E staining slides of the liver tissue from wild type, whole body PPAR α KO and liver-specific PPAR α KO mice, which were fed with a HFHS diet for 12 weeks. Severe hepatic steatosis was present in all of these mice and centered on the central veins with the steatosis score of 3. This is consistent with previous report that the whole body and liver-specific PPAR α KO mice developed steatosis around the central veins and high fat diet (HFD) further promoted liver steatosis (PMID: 32300166). Thus, better representative images of liver H&E staining have been shown in the **revised Fig. 2d** and **new Fig.5 b and j**.

5. It's also weird to show all the relative gene/protein expression level in the control/vehicle group as 100.

Response: The data of gene/protein expression in model or treated groups are usually visualized as relative expression to control group, which is set as 100% or 1

(PMID:36477534, Fig1b; PMID: 36544023, Fig2b; PMID: 37208332, Fig 6b; PMID: 31672964, Fig 6a and b; PMID: 33338411, Fig 4g, et al). In current study, the average of gene/protein expression in control group is normalized as 100%. We have clarified the method for these data analysis in the figure legends.

Reviewer #3 (Remarks to the Author):

In this manuscript, Zhong et al. found that the serum HDCA significantly reduced in NAFLD patients, and HDCA showed strongly inverse correlation with clinical parameters of NAFLD. Meanwhile, it was verified that HDCA can improve diet-induced NAFLD by animal experiments. The mechanism by which HDCA ameliorated diet-induced NAFLD is that HDCA hindered the formation of RAN/CRM1/PPAR α shuttle heterotrimer by direct binding with RAN protein, thereby leading to the accumulation of nuclear PPAR α . This study uncovered a novel mechanism underlying the anti-NAFLD effect of HDCA and provide the rationale for novel therapeutic strategy on NAFLD. Overall, the work is well planned and the findings are supported by the results and this manuscript is well organized with good logic. Therefore, it could be considered for publication after addressing the following issues.

1. The information of the recruited patients is not comprehensive. Are they taking drugs for the treatment of NAFLD? Such as hepatoprotectants, hypoglycemic and lipid-lowering drugs? and will these drugs affect the research results?

Response: Thanks for the reviewer's suggestion. Specific inclusion and exclusion criteria are set out in the **revised supplementary document**. We further refined the clinical information of NAFLD patients and healthy controls, especially the medication use information (**new Table S1**). In detail, none of the healthy controls took any medication, while among the 34 NAFLD patients, 2 patients (NO. 3, 11) took both antihypertensive and hepatoprotective drugs, 1 patient (NO. 21) took antihypertensive drugs, 1 patient (NO. 20) took both antihypertensive and non-beta and non-statin lipid-lowering drugs, and 1 patient took hepatoprotective drugs (NO.12). OPLS-DA modeling profile of bile acids in NAFLD patients and healthy controls showed no abnormal deviation of these patients. In addition, even when the patients who took medicines were excluded, the differences in HCA species and HDCA between healthy controls and NAFLD patients were not altered (**Response to reviewer Fig. 4**), implying that the medication in NAFLD patients did not affect the bile acid profile. The criteria for patient enrollment and exclusion were revised in the Supplementary document.

Response to reviewer Figure 4. Serum bile acid profile of health and NAFLD individuals. (A) OPLSDA plot of serum BA profile in health ($n=24$) and NAFLD ($n=34$) individuals. (B) Serum level of HCA species, HDCA, GHDCa, HCA, GHCA and THCA of health ($n=24$) and NAFLD ($n=29$) individuals without taking any medication.

2. To verify that the anti-NAFLD effect of HDCA is PPAR α -dependent, it is recommended that the antiNAFLD effect of HDCA should be tested by liver-specific Ppara knockout mice in animal experiments.

Response: We appreciate this important suggestion. In the revised manuscript, the liver-specific PPAR α knockout mice were generated by administering *Ppara*^{flox/flox} mice with AAV2/8-TBG-Cre (**new Fig. S14d**). Similar to the response of global PPAR α KO mice (**new Fig. 5a-h**), HDCA had no effect on reducing liver TG, ameliorating hepatic steatosis, improving inflammation, or regulating targeted protein expression in liver-specific PPAR α KO mice, suggesting the anti-NAFLD effect of HDCA relies on liver PPAR α (**new Fig. 5i-o**). These data have been included in the result and discussion sections (**new Fig. 5, new Fig. S14**, Page 9, Line 216-226; Page 15, Line 363-368; Page 19, Line 461-469).

3. In animal experiments 1-4 (SUPPLEMENTARY METHODS AND MATERIALS), Why are the modeling time and dose of HDCA inconsistent? Is there any literature support for experimental scheme? Please add references. In animal experiment 3, The grouping and experimental scheme of HDCA group are not clearly described (line 47).

Response: Thanks for pointing out this important question. There are 5 animal experiments in our current study. The experimental scheme of these animal experiments is explained as following:

Animal experiment 1: Actually, the time points for inducing NAFLD with HFD or HFHS diet varied greatly from 8 to 24 weeks according to most publications (PMID:27261415; PMID: 35928832). In our current study, we first conducted a preliminary observational experiment by feeding C57 mice with HFHS for 4, 8, 12 and 24 weeks, respectively, in which the extent of hepatic steatosis was evaluated based on liver histology with H&E staining. We found that 8-week HFHS feeding resulted in the presence of hepatic

steatosis, and the extent of hepatic steatosis progressed time-dependently. Obviously, fatty liver was well-established in mice by 24-week HFHS feeding (**Response to reviewer Figure 1**), which is consistent with previous reports (PMID:27261415). Based on these observations, a 24-week HFHS feeding design was first used for generating a typical NAFLD mouse model in C57 mice, and samples from these mice were subjected for identifying NAFLD-related bile acids.

Animal experiment 2: As for the dose and time points used in HDCA intervention experiment, we have explained the rationale for dose determination of HDCA in **Response to Question 3 of Reviewer 2** above, and similar **Response to Question 3 of Reviewer 1** on the time points selection. We chose 12 weeks for subsequent evaluation on the anti-NAFLD effect of HDCA is due to the observation of HFHS diet impact on hepatic steatosis formation from 4 to 24 weeks. It is obvious that 12 weeks HFHS feeding is sufficient for inducing successful hepatic steatosis (**Fig. 2d-e**), which is more applicable for evaluating the anti-NAFLD effect of HDCA in mice compared to the long term 24 weeks feeding.

Animal experiment 3: Since *ob/ob* mice are easier to develop obesity and NAFLD under either normal chow diet or HFD than wild type C57 mice (PMID:21740407, 25484077), the experiment on *ob/ob* mice was terminated at the end of 8 weeks when the body weight of *ob/ob* mouse was almost 60 g averagely. We think 8 weeks of HFHS feeding is sufficient for inducing NAFLD in *ob/ob* mice, which is well confirmed in Fig S7. Meanwhile, given the probable more severe extent of hepatic steatosis in *ob/ob* mice than wild type C57 mice, we think it is necessary to increase the dose of HDCA in *ob/ob* mice. So we used the dose of HDCA at 1.25% supplemented in diet, which was equal to the high dose trialed in wild type C57 mice.

Experiment 4 and 5: Since the main purpose of experiment 4 and 5 was to determine whether the anti-NAFLD effect of HDCA was PPAR α , especially liver PPAR α dependent, we therefore used the dose of HDCA and experiment period that was identical with wild type mice.

We have added the relative description in the Methods section in this revision (Page 17 Line 426-Page19 Line 469).

4. Some notes (e.g. Figure S1, and S6) are not corresponding to the right figures, which should be double-checked and revised.

Response: Thanks a lot for pointing out these issues. We have double-checked and revised them.

5. Several images on KEGG enrichment analysis and heatmap of proteins related to fatty acid metabolism are blurry. It is suggested to improve the resolution of pictures in the article.

Response: Thanks, we have improved the resolution of these images in the revised manuscript.

REVIEWERS' COMMENTS

Reviewer #1 (Remarks to the Author):

The Authors have addressed most of the reviewers concerns.

Reviewer #2 (Remarks to the Author):

I have no further comments.

Reviewer #3 (Remarks to the Author):

Authors have tried their best to answer my questions. I think that all of my concerns have been well addressed in this revision by adding sufficient new evidence. The main results supported the conclusions of the study, and the current findings are novel for understanding the mechanism of HDCA in NAFLD treatment. Therefore, the revised manuscript is now acceptable for NC publication.

REVIEWERS' COMMENTS

Reviewer #1 (Remarks to the Author):

The Authors have addressed most of the reviewers concerns.

Reply: Thank you very much for your valuable comments and suggestions on our work.

Reviewer #2 (Remarks to the Author):

I have no further comments.

Reply: Thank you very much for your valuable comments and suggestions on our work.

Reviewer #3 (Remarks to the Author):

Authors have tried their best to answer my questions. I think that all of my concerns have been well addressed in this revision by adding sufficient new evidence. The main results supported the conclusions of the study, and the current findings are novel for understanding the mechanism of HDCA in NAFLD treatment. Therefore, the revised manuscript is now acceptable for NC publication.

Reply: Thank you very much for your valuable comments and suggestions on our work.